# Effect of Dietary Silk Peptide on Obesity, Hyperglycemia, and Skeletal Muscle Regeneration in High-Fat Diet-Fed Mice

**DOI:** 10.3390/cells9020377

**Published:** 2020-02-06

**Authors:** Kippeum Lee, Heegu Jin, Sungwoo Chei, Hyun-Ji Oh, Jeong-Yong Lee, Boo-Yong Lee

**Affiliations:** 1Department of Food Science and Biotechnology, College of Life Science, CHA University, Seongnam, Kyonggi-do 13488, Korea; joy4917@hanmail.net (K.L.); heegu94@hanmail.net (H.J.); sungwoochei@gmail.com (S.C.); guswl264@naver.com (H.-J.O.); 2Worldway Co., Ltd., Sanda-gil, Jeonul-myeon, Sejong-si 30003, Korea; jy.lee@worldway.co.kr

**Keywords:** diabetes, muscle differentiation, obesity, sarcopenia, silk peptide

## Abstract

Obesity is associated with excess body fat accumulation that can cause hyperglycemia and reduce skeletal muscle function and strength, which characterize the development of sarcopenic obesity. In this study, we aimed to determine the mechanism whereby acid-hydrolyzed silk peptide (SP) prevents high-fat diet (HFD)-induced obesity and whether it regulates glucose uptake and muscle differentiation using in vivo and in vitro approaches. Our findings demonstrate that SP inhibits body mass gain and the expression of adipogenic transcription factors in visceral adipose tissue (VAT). SP also had an anti-diabetic effect in VAT and skeletal muscle because it upregulated glucose transporter type 4 (GLUT4) and uncoupling protein 3 (UCP3) expression. Furthermore, SP reduced ubiquitin proteasome and promoted myoblast determination protein 1 (MyoD)/myogenic factor 4 (myogenin) expression, implying that it may have potential for the treatment of obesity-induced hyperglycemia and obesity-associated sarcopenia.

## 1. Introduction

Obesity is a global health concern and is characterized by an expansion of adipose tissue, which reflects an imbalance between energy intake and expenditure [1,2]. It is associated with a number of metabolic diseases, such as type 2 diabetes (T2D), cardiovascular disease, hypertension, stroke, chronic low-level inflammation, and various cancers [3,4]. Adipose tissue is an important metabolic organ, which expands in terms of cell size and number to permit greater lipid storage during long-term positive energy imbalance [5]. Abdominal white adipose tissue (WAT) fat has a vital role in the regulation of systemic energy homeostasis [6], and can be classified as visceral (VAT) and subcutaneous (SAT) [7]. If excess lipid principally accumulates in VAT, which is the more metabolically active depot, glucose intolerance is more likely to develop than if this accumulation occurs in SAT. Therefore, the measurement of VAT volume is thought to be of more metabolic significance than the assessment of general abdominal obesity [8].

Adipogenesis is the process whereby pre-adipocytes differentiate into mature adipocytes, which is required for the storage of excessive amounts of lipid in adipose tissue [9]. Multiple transcription factors regulate this process, including CCAAT/enhancer-binding protein alpha (C/EBPα), and peroxisome proliferator-activated receptor gamma (PPARγ). The expression of all of these transcription factors is required for adipocyte differentiation and the expression of adipocyte-specific genes [10,11].

Diabetes is one of the most important current public health problems worldwide [12]. It is associated with obesity, aging, and impaired cognitive performance, and is characterized by hyperglycemia and hyperinsulinemia, which reflect glucose intolerance and insulin resistance [13]. Hyperglycemia develops as a result of an impairment in glucose uptake into skeletal muscle and WAT, secondary to a defect in the action of insulin, the key regulator of glucose homeostasis. Such insulin resistance can often be identified in obese, as well as diabetic, patients [14], and underpins metabolic syndrome, which is a cluster of cardiovascular risk factors that includes obesity [15,16]. Skeletal muscle is the primary tissue for glucose uptake, which occurs via glucose transporter 4 (GLUT4), and it is a rate-limiting step in insulin-dependent glucose utilization [17,18]. Numerous studies have shown that insulin stimulates glucose uptake into peripheral tissues through activation of the insulin receptor, which phosphorylates insulin receptor substrate 1 (IRS1). This adaptor protein activates the downstream protein kinase B (AKT), which in turn induces the translocation of GLUT4, thus increasing glucose uptake and reducing the circulating glucose concentration [19,20].

Sarcopenia is defined as the degenerative loss of skeletal muscle and strength that occurs with aging [21]. Although the etiology and pathogenesis of sarcopenia have not been fully established, sarcopenia has been suggested to play an important role in the clinical syndrome associated with diabetes. T2D has been reported to be an important risk factor for the development of physical disability in older adults because of their lower muscle mass and poorer muscle strength [22,23]. Furthermore, recent studies have shown that obesity and the redistribution of body fat to VAT is also associated with skeletal muscle loss and dysfunction, in a syndrome referred to as sarcopenic obesity [24]. E3 ubiquitin ligases, such as F-box protein (Fbx32/atrogin) and muscle ring finger1 (MuRF1), which are involved in the ubiquitin proteasome pathway, are thought to induce protein degradation in sarcopenia [25].

Skeletal muscle makes the largest contribution to total body energy expenditure [26]. Various signal transduction systems in this tissue can affect the activation of peroxisome proliferator-activated receptor γ coactivator 1α (PGC1α) and nuclear respiratory factor 1 (NRF1), which enhance mitochondrial biogenesis and increase the expression of uncoupling protein 3 (UCP3). UCP3, which is expressed in the mitochondrial inner membrane, uncouples the oxidation of fuels from ATP production, which results in the loss of energy as heat. Therefore, the induction of UCP3 secondary to an upregulation of mitochondrial biogenesis in muscle has been hypothesized as a potential means for energy homeostasis [27,28].

Silk peptide (SP), derived from *Bombyx mori* cocoons, is an edible biomaterial that has a variety of applications in the food, cosmetics, and biotechnology industries in Asian countries [29,30], but is frequently used as a food supplement. A recent study showed that silk fibroin enhances insulin sensitivity and glucose uptake in 3T3-L1 adipocytes [31]. In addition, it has been shown that SP increases fat oxidation in exercising mice [32]. Finally, in our previous study, we demonstrated that SP has an anti-obesity effect by promoting beige-like adipocyte differentiation in WAT via AMP-activated protein kinase (AMPK) activation in high-fat diet (HFD)-induced obese mice [33]. However, the mechanisms of its effects on glucose homeostasis and whether it affects muscle differentiation have not been determined in vivo. Therefore, in this study, we aimed to determine the effect of SP on lipid and glucose metabolism and muscle differentiation in HFD-fed mice.

## 2. Materials and Methods

### 2.1. Materials

The dietary acid-hydrolyzed silk peptide (SP) used in this study was prepared from the cocoons of *Bombyx mori*, and is an ingredient for manufacturing commercial products used by Worldway Co., Ltd. (Sejong, Korea, Lot number, 1803002). In brief, raw cocoons were acid-hydrolyzed, and the resulting solution was neutralized, decolorized, filtered, desalted, and freeze-dried to produce a pale yellow powder. The mean molecular weight of the acid-hydrolyzed SP ranged from 150 to 300, and it was analyzed by mass spectrometry, as described previously [33].

Fetal bovine serum (FBS) and horse serum were purchased from Gibco (Gaithersburg, MD, USA) and bovine serum (BS) was purchased from Corning (Corning, NY, USA). An enzyme-linked immunosorbent assay (ELISA) for HbA1c was purchased from Merck Millipore (Temecula, CA, USA) and ELISA kits for triglycerides and total cholesterol were purchased from Abcam (Abcam, Cambridge, UK). Anti-carnitine palmitoyltransferase 1 (CPT1, ab128568), GLUT4 (ab35826), MuRF1 (ab172479), Myosin/MYH3 (ab124205), NRF1 (ab175932), PPARα (ab24509), PRDM16 (ab202344), Fbx32 (ab168372), and UCP3 (ab10985) were purchased from Abcam. Anti-PGC1α (sc13067), AKT (cs9272), p-AKT (Ser 473, cs9271), AMPK (cs2532), and p-AMPK (Thr 172, cs2535) antibodies were obtained from Cell Signaling Technology (Danvers, MS, USA). Anti-C/EBPα (sc61), PPARγ (sc7273), myogenin (sc12732), DGAT1 (sc32861), MyoD (sc760), GAPDH (sc365062), IRS (sc559), and p-IRS (Tyr 632, sc17196) antibodies were purchased from Santa Cruz Biotechnology(Santa cruz, CA, USA). Metformin was purchased from Sigma Aldrich (St. Louis, MO, USA).

### 2.2. Animals and Experimental Design

Male 4-week-old ICR (CrljOri:CD1) mice were purchased from Joong-Ah Bio (Suwon, Korea). The animal experiments were approved by the Institutional Animal Care and Use Committee (IACUC) of CHA University (IACUC approval number, 190173). The mice were initially housed for 1 week under a 12 h light/dark cycle condition at a 20–24 °C temperature and 44–52% humidity to permit adaptation. After adaptation, the mice were randomly allocated to four groups (*n* = 8 per group) and then fed for 6 weeks with a chow diet (CD, 10% of calories derived from fat; D12450B, Research Diets, NJ, USA) or an HFD (60% of calories derived from fat; D12492, Research Diets). Over the same period, SP (50 and 200 mg/kg/day) or an equal volume of vehicle was orally administered daily to the mice consuming HFD. The SP doses administered to the mice were derived from the human doses (0.25 g/60 kg/day and 1 g/60 kg/day) using a mathematical table, as previously described [34]. The body mass, food intake, and water consumption were recorded weekly. At the end of the experimental period, the mice were fasted for 12 h and euthanized using the gradual-fill method of carbon dioxide euthanasia, and their tissues were collected for analysis. The organs were weighed carefully.

Six-week-old male C57BL/6J mice (YM) and 12-month-old male C57BL/6J mice (AM) were purchased from Joong-Ah Bio, and housed under the same conditions as those of ICR mice. All mice were fed a CD and watered libitum during the experiment. After 2 weeks of adaptation, the YM group (*n* = 8) and AM group were divided into two groups of equal sizes (*n* = 8). The AM + SP250 group was orally given SP dissolved in water at a dose of 250 mg/kg/day for 8 weeks. YM and AM groups were given an equal volume of water to that provided in the OM + SP250. The body weight of the mice was recorded at 0, 4, and 8 weeks.

### 2.3. Fasting Blood Glucose Measurement

Fasting blood glucose was measured weekly in blood obtained from a tail vein after withholding food for 12 h, using a glucose analyzer (GlucoDr, Allmedicus, Kyeonggi, Korea).

### 2.4. Oral Glucose Tolerance Testing

Oral glucose tolerance testing (OGTT) was performed after 5 weeks of treatment after the mice had been fasted for 12 h. Each mouse was administered 1.5 g/kg body mass D-glucose orally, and then every 30 min, the glucose concentration was measured in a blood sample collected from the tail vein, as described above.

### 2.5. Measurement of Rectal Temperature

At the end of the experimental period, the rectal temperatures of the mice were measured four times using a Testo 925 Type Thermometer (Testo, Lenzkirch, Germany).

### 2.6. Grip Strength Test

The grip strength of the mice was assessed using a grip strength meter with a single sensor, which is called the Chatillon force measurement system (Columbus Instrument, OH, USA). The mice were placed with their forelimbs and hindlimb on a narrow bar and the strength when mice fell was measured at the end of the oral administration period.

### 2.7. Serum Biochemical Analyses

Blood was collected by cardiac puncture at the time of euthanasia. Serum was then separated by centrifugation at 6000× *g* for 10 min at 4 °C, after clotting. The triglyceride, total cholesterol, - HbA1c, creatinine, aspartate aminotransferase (AST), and alanine aminotransferase (ALT) concentrations were then measured by ELISA.

### 2.8. Histologic and Immunofluorescence Analyses

WAT and gastrocnemius muscle in hindlimb samples were rapidly collected and fixed in 4% paraformaldehyde solution for 48 h. The tissues were then paraffin-embedded and the resulting blocks were cut into 10 µm sections and stained with hematoxylin and eosin (H&E) to assess the histology. Photomicrographs were obtained using a Nikon Eclipse E600 microscope (Nikon Corporation, Tokyo, Japan).

For immunofluorescence analysis, the tissues were fixed and stained with rabbit-GLUT4 (dilution, 1:500), -UCP3 (dilution, 1:200), or -MYH3 (dilution, 1:1000) antibodies overnight at 4 °C in a moist chamber. Alexa Fluor™ 594-conjugated and fluorescein isothiocyanate (FITC)-conjugated (dilution, 1:500) secondary antibodies were used, and DAPI (Sigma Aldrich, St. Louis, USA) was used to stain the cell nuclei. After mounting using ProLong Gold Antifade reagent (Thermo Fisher Scientific, Waltham, MA, USA), fluorescent images were captured using a Zeiss confocal laser scanning microscope (LSM880; Carl Zeiss, Oberkochen, Germany) and Zen 2012 software (Carl Zeiss).

### 2.9. Microcomputed Tomography (Micro-CT) Analysis

The mice were intraperitoneally anesthetized with 1 mL/kg zoletil:rompun solution (4:1, voleme/volume) at 0, 4, and 8 weeks. Hindlimb muscle of the mice was obtained by Micro-CT analysis with a single photon emission tomography system at the research center of Chemon Inc. (Gyeonggi-do, Korea). Total hindlimb muscle volumes were obtained using Siemens Inveon software (Inveon, Washington, DC, USA).

### 2.10. Forced Swimming Test (FST)

Mice underwent pre-swim training twice, one week before the FST. Mice were placed in cylinders containing 30 cm of clean water at 23 ± 2 °C for 10 min. The mice were fasted for 4 h before the test and weighed upwards of 5% body mass for the tail, and were then placed in the water-filled cylinder. The trial lasted 30 min and its latency was recorded.

### 2.11. Cell Culture and Treatment

C2C12 myoblasts were maintained in growth medium (DMEM medium containing 10% FBS, 1% penicillin/streptomycin (P/S), and 3.7 g/L sodium bicarbonate) in a humidified 5% CO_2_ incubator at 37 ℃. The cells were passaged by trypsinization at 60% confluence. To differentiate the myoblasts, 80–90% confluent cells were incubated in DMEM containing 2% horse serum for 5 days, which was refreshed daily.

3T3-L1 pre-adipocytes were cultured in growth medium (DMEM containing 10% BS, 1% P/S, and 3.7 g/L sodium bicarbonate) at 37 °C in a humidified 5% CO_2_ incubator to confluence, with the medium being replaced every 2 d. Two days after becoming 100% confluent, the 3T3-L1s were induced to differentiate in DMEM medium containing 10% FBS, 1 mM dexamethasone, 0.5 mM 3-isobutyl-1-methylxanthine, and 5 mg/mL insulin, which was then refreshed every 2 d with DMEM supplemented with 10% FBS and 5 mg/mL insulin. SP was prepared as a 400 mM stock solution in distilled water and then diluted in medium to final concentrations of 25, 50, 100, 200, or 400 μM. Metformin (2 mM) was used as a positive control treatment.

### 2.12. Cell Viability Test

Cells were seeded (3 × 10^3^ cells/well) in 96-well plates and incubated overnight in growth medium. Cells were then treated with SP (0, 12.5, 25, 50, 100, or 200 μg/mL) and incubated for a further 24 h. Next, 20 μL 3-(4,5-dimethyl-2-thiazolyl)-2,5-diphenyl-2H-tetrazolium bromide (MTT) solution was added to each well and the cells were incubated for a further 4 h. The MTT-containing medium was then removed and 100 μL DMSO was added to elute the formazan crystals. The absorbances of the eluates were measured at 570 nm (BioTek, Winooski, VT, USA).

### 2.13. Oil Red O Staining

After 8 days of differentiation of the 3T3-L1 cells, they were fixed in 10% formaldehyde for 2 h at room temperature and then washed twice with phosphate-buffered saline (PBS). The cells were stained with Oil red O (ORO) in a 6:4 (*v*/*v*) solution of isopropanol:distilled water for 40 min. After washing and drying, the stained cells were imaged.

### 2.14. Western Blot Analysis

Tissues or cells were lysed in lysis buffer (iNtRON Biotechnology, Seoul, Korea) containing phosphatase and protease inhibitors. The concentration of protein in each lysate was then quantified using a protein assay kit (Bio-Rad, Hercules, CA, USA). Equal amounts of protein (20 μg) were diluted in 5× sample buffer (50 mM Tris pH 6.8, 2% sodium dodecyl sulfate (SDS), 10% glycerol, 5% β-mercaptoethanol, and 0.1% bromophenol blue) and heated for 5 min at 90 °C. These samples were separated using 8%–12% SDS-polyacrylamide gel electrophoresis and transferred to polyvinylidene fluoride membranes. The membranes were blocked using 5% non-fat dried milk for 1 h and then incubated overnight at 4 °C with primary antibodies diluted to 1:1000. After washing, the membranes were then incubated with secondary antibodies conjugated to horseradish peroxidase (1:5000), and specific protein bands were detected by enhanced chemiluminescence and then imaged using an Amersham Imager 680 (GE Healthcare Life Sciences, Chicago, IL, USA).

### 2.15. Statistical Analysis

The data were analyzed using one-way ANOVA and Duncan’s test (SPSS, ver 6.0 Chicago, IL, USA) and are presented as the mean ± SD. Statistical significance was accepted when *p* < 0.05. The supplementary data analyzed using Student’s *t*-test (SPSS, ver 6.0 Chicago, IL, USA) and * *p* < 0.05, ** *p* < 0.05 were compared YM at 0 week, and ^#^
*p* < 0.05, ^##^
*p* < 0.05 were compared AM at 0 week.

## 3. Results

### 3.1. SP Ameliorates Body Mass Gain in HFD-Fed Mice

To evaluate the anti-obesity effects of SP in vivo, we used an HFD-induced obese ICR mouse model. After 6 weeks, the body mass of control HFD-fed mice was significantly higher than that of CD-fed mice, but was reduced by SP administration in a dose-dependent manner (Figure 1A,B). The body mass gain of HFD-fed mice was 23.9 ± 1.7 g, but that of the SP50 and SP200 groups was 19.3 ± 3.9 g and 13.5 ± 3.8 g, respectively. The mean VAT mass was effectively reduced by 16.4% and 49.7% by SP administration at 50 and 200 mg/kg/day, respectively (Figure 1C). Additionally, SAT mass was significantly reduced by SP treatment in a dose-dependent fashion. By contrast, there were no differences in the masses of BAT, liver, lung, spleen, or kidney among the groups (Table 1). Representative images of mice from each group are shown as Figure 1D. Visceral obesity is known to be associated with high serum cholesterol and triglyceride concentrations [35]. As shown in Figure 1E–F, the serum total cholesterol concentration was 134.0 ± 4.5 mg/dL in the HFD group, but only 117.5 ± 6.8 mg/dL and 113.3 ± 3.9 mg/dL in the SP50 and SP200 groups, respectively. Likewise, the triglyceride concentration in the HFD group was 115.3 ± 5.2 mg/dL, and that of the SP50 and SP200 groups was 101.5 ± 6.6 mg/dL and 97.3 ± 7.5 mg/dL, respectively. There were no differences in food and water intake among the groups, suggesting that the inhibition of body mass gain in SP-treated mice was not due to a lower food intake. Lastly, to investigate the effect of SP on energy expenditure, the rectal temperature of the mice was measured at the end of the administration period. As a result, SP significantly increased the body temperature to 38.0 ± 0.4 and 38.0 ± 0.3 °C at 50 and 200 of the SP-treated group, respectively. Furthermore, Table 2 indicates that SP displays no significant differences in serum creatine or AST and ALT concentrations, which are markers of renal injury or indicative of liver damage, respectively. Therefore, SP treatment does not affect hyperthermia by systemic inflammation associated with acute kidney or liver injury.

### 3.2. SP Regulates the Blood Glucose Concentration in HFD-Fed Mice

Hyperglycemia is associated with obesity and can progress to pre-diabetes, which is characterized by an impaired glucose tolerance and insulin resistance [36]. Therefore, we determined whether the HFD induced hyperglycemia in the mice and whether SP was able to prevent this. As shown in Figure 2A–B, the fasting blood glucose concentrations in all the mouse groups before treatment were within the normal range (106.7 ± 2.9 mg/dL). However, the blood glucose concentration in HFD-fed mice had markedly increased after 2 weeks of diet consumption to 162.8 ± 8.3 mg/dL. By contrast, SP administration prevented this increase in a dose-dependent manner. OGTT was then used to evaluate the effect of SP ingestion on glucose tolerance (Figure 2C,D). After the administration of 1.5 g/kg glucose to the mice, the blood glucose of the CD-fed group was 196.2 ± 21.5 mg/dL at 30 min, but that of the HFD-fed group was 289.7 ± 12.5 mg/dL. Compared with the blood glucose of the HFD-fed mice at 60 min, the groups that were administered 50 or 200 mg/kg/day SP had significantly lower concentrations, by approximately 14.9% and 29.2%, respectively. At 120 min, the blood glucose of the HFD-fed group had decreased to 129.2 ± 9.6 mg/dL, and the group that received 200 mg/kg/day SP had a blood glucose concentration that was only a little lower than this (111 ± 19.2 mg/dL). T2D is characterized by high serum hemoglobin A1c (HbA1c) [37]. As shown in Figure 2E, the HbA1c of HFD-fed mice was significantly higher than that of the CD-fed mice and the SP-administered groups.

### 3.3. SP Ingestion Reduces Adiposity in HFD-Fed Mice

To determine whether SP administration has an anti-obesity effect, we analyzed adipose tissue from the experimental mice and SP-treated 3T3-L1 adipocytes. Sections through the VAT depots of mice from each group were stained using H&E and examined microscopically. As shown in Figure 3A,B, the adipocyte size in HFD-fed mice was much larger than in CD-fed mice, but SP50 and SP200 prevented this increase. To interrogate the mechanisms of the effect of SP on adipose tissue, the expression of key regulators of adipogenesis, C/EBPα and PPARγ, was analyzed using western blotting. This showed that C/EBPα and PPARγ protein expression was 354% and 234% higher in the VAT of mice fed an HFD for 6 weeks than in CD-fed mice, respectively. However, the expression of C/EBPα and PPARγ in both SP-treated groups was lower than in HFD-fed mice (Figure 3C). Adiponectin is an adipokine that plays a vital role in preventing the development of obesity, T2D, and metabolic syndrome [38]. The administration of SP increased the expression of adiponectin in VAT.

SP has previously been shown to inhibit lipid metabolism in the 3T3-L1 cell line [39]. To evaluate the cytotoxicity of SP in this cell line, an MTT assay was performed, and no cytotoxicity was identified at concentrations up to 400 µg/mL. ORO staining of 3T3-L1 adipocytes treated with 25–100 µg/mL SP showed that lipid accumulation in these cells was substantially prevented by SP, as shown in Figure 3E. However, 200 µg/mL SP induced fat accumulation. Consistent with this, SP reduced the expression of the adipogenic markers C/EBPα and PPARγ when administered at concentrations up to 100 µg/mL, but 200 µg/mL SP increased the expression of these proteins. This implies that lower levels of consumption of SP may result in lower weight gain by reducing adipose tissue development, but an excessive intake of SP should be avoided.

### 3.4. SP Increases Glucose Uptake into Adipose Tissue

To determine the mechanism whereby SP ameliorates hyperglycemia, we measured the expression and phosphorylation of insulin-signaling intermediates and GLUT4. The protein expression of IRS1, p-IRS1, AKT, p-AKT, and GLUT4 in the VAT of each group of mice is illustrated in Figure 4A. The CD-fed mice had lower p-IRS1 and p-AKT expressions than the HFD-fed mice, and SP markedly increased the phosphorylation of IRS1 and AKT in VAT. Compared with the HFD-fed mice, the 200 mg/kg/day SP group showed significant increases in IRS-1 and AKT phosphorylation of 273% and 298%, respectively. In addition, 200 mg/kg/day SP administration increased GLUT4 expression in the VAT by 242% *versus* that in the HFD-fed mice. GLUT4, the insulin-responsive glucose transporter, is expressed in adipose tissue and gastrocnemius muscle in the hindlimb, and plays a critical role in glucose homeostasis [40]. Moreover, it is known that typical western diets, which include a high-fat content, are major factors in the induction of obesity and insulin resistance.

Likewise, during adipogenesis in 3T3-L1cells, SP treatment increased IRS1 phosphorylation, AKT phosphorylation, and GLUT4 expression. As shown in Figure 4B, the western blotting data imply that SP increased the activation of the insulin signaling pathway in a dose-dependent manner. Furthermore, compared with 3T3-L1s treated with Met, a drug that is widely used for the treatment of T2D, the expression levels of p-IRS and p-AKT in cells treated with 100 μg/mL SP were higher. In addition, cells treated with >50 μg/mL SP expressed GLUT4 at similar levels to Met-treated 3T3-L1. These data suggest that SP may promote glucose uptake in HFD-induced obese mice by activating the insulin signaling pathway and increasing GLUT4 expression.

### 3.5. SP Increases the Phosphorylation of Insulin Signaling Intermediates and Mediators of Energy Consumption in Skeletal Muscle

Skeletal muscle is one of the main tissue regulators of glucose metabolism, which makes normal muscle metabolism important for whole-body glucose homeostasis. In particular, a substantial increment in muscle glucose uptake results from the translocation of GLUT4 to the sarcolemma in response to insulin [41]. However, in addition, AMP-activated protein kinase (AMPK), an energy sensor that is important in metabolic homeostasis, also stimulates glucose uptake and mitochondrial biogenesis, and has been shown to ameliorate insulin resistance in this way [42]. In the present study, HFD-fed mice showed 28.4% lower AMPK phosphorylation in skeletal muscle than CD-fed mice, but SP treatment normalized this. In addition, p-IRS expression was higher in the SP-administered groups than in CD- and HFD-fed mice (Figure 5A). In skeletal muscle, GLUT4 expression was significantly lower in HFD-induced obese mice. However, SP50 and SP200 treatment markedly increased GLUT4 expression.

We next evaluated the impact of HFD-feeding and SP treatment on muscle PGC1α, NRF1, and UCP3 expression in skeletal muscle (Figure 5B). Although there were no significant differences in PGC1α expression among the groups, NRF1 expression in SP200 mice was much higher than in the other groups. In addition, our data show that SP administration significantly increased the expression of UCP3 in a dose-dependent fashion, such that it was higher than that of CD-fed mice. Consistent with the above findings, immunofluorescence demonstrated more intense staining for GLUT4 and UCP3 expression in the skeletal muscle of SP-treated groups than in the HFD-fed group, as shown in Figure 5C. These findings imply that both skeletal muscle glucose uptake and mitochondrial energy expenditure are increased by SP treatment.

### 3.6. SP Increases the Expression of Genes Determining Skeletal Muscle Regeneration and Reduces Those Involved in Sarcopenia

Muscle comprises >40% of the body mass, and the maintenance of skeletal muscle mass and strength is considered essential for maximizing the life span [43]. Interestingly, when the grip strength was measured to assess the muscle exercise capacity, it was found to be 29.9% lower in HFD-fed mice than in CD-fed mice, but this defect was ameliorated in SP-administered HFD-fed groups (Figure 6A). In addition, H&E-stained images of gastrocnemius muscle in the hindlimb (Figure 6B,C) established that SP treatment dose-dependently increased the cross-sectional area (CSA), which implies that SP has an effect on the size of muscle fibers.

This change in gastrocnemius muscle in hindlimb morphology seemed to be correlated with a lower expression of muscle differentiation factors in HFD-fed mice. Indeed, 200 mg/kg/day SP administration significantly increased myogenin to levels similar to those in CD-fed mice, and the MyoD expression in mice treated with 200 mg/kg/day SP was higher than that of CD-fed mice (Figure 6D). We also measured the expression of Fbx32 and MuRF1, E3 ubiquitin ligases that are upregulated in skeletal muscle undergoing sarcopenia [44]. The expression of Fbx32 and MuRF1 in SP200 mice was 70% and 76% lower, respectively, than that of HFD-fed mice. Therefore, SP may prevent sarcopenia by enhancing muscle differentiation and inhibiting the muscle atrophy induced by HFD-feeding.

A reduction in muscle mass and an increase in body weight are some of the most striking changes associated with advancing age. These changes can have metabolic dysfunctional consequences in the body [45]. In this study, we investigated the effect of SP on aging-induced sarcopenia end exercise using the 6-week-old and 12-month-old aged mice. As shown in Figure 7A,B, the total hindlimb muscle of the AM group was significantly reduced during 8 weeks compared with that of YM. However, the muscle volume of 250 mg/kg/day SP administrated AM (AM + SP250) was effectively increased for 8 weeks. In addition, FST data shows that the exercise time was longer with SP treatment in AM, which implies that SP enhances the muscle strength by increasing the muscle mass (Figure 7C). However, the body weight of AM + SP250 was not significantly different from that of the AM group, but slightly decreased (Figure 7D).

### 3.7. SP Stimulates Myoblast Differentiation and Glucose Metabolism in C2C12 Cells

C2C12 derived from murine skeletal muscle is a well-known cell line used to determine muscle regeneration and differentiation [46]. We next evaluated the effects of SP on myoblast differentiation and glucose uptake in the C2C12 cell line. We first evaluated the viability of cells treated with various concentrations of SP, and found that it was cytotoxic at concentrations >200 μg/mL (Figure 8A). C2C12 cells were then incubated in differentiation medium containing various concentrations of SP or 2 mM Met for 5 days. Each group was observed by an optical microscope, as shown in Figure 8B. Treatment with SP significantly increased the expression of MyoD, Myogenin, and MYH3 in C2C12 cells compared with untreated cells (Figure 8C). Met-treated C2C12s were used as a positive control and also exhibited a higher expression of these genes, but this was slightly lower than that of the SP-treated cells. In addition, higher expression levels of MyoD, Myogenin, and MYH3 were found in cells treated with 25–100 μg/mL SP, depending on the concentration (Figure 8C). Therefore, SP appears to promote the differentiation of myoblasts. Moreover, as shown in Figure 8D, 100 µg/mL SP increased p-AMPK, NRF1, UCP3, and GLUT4, implying that SP upregulates mitochondrial activity and glucose uptake. Met treatment also increased the expression of these proteins, but the GLUT4 expression level in Met-treated cells was lower than that in all SP-treated cell groups. These results imply that SP may stimulate glucose uptake in myoblasts by increasing mitochondrial metabolism and GLUT4 expression.

## 4. Discussion

SP is a bioactive material that is extracted from the silkworm *Bombyx mori* and has a potential value in traditional medicine and as a dietary supplement for a broad range of health-related purposes [47]. Recent research has shown that SP has anti-diabetic, anti-tumor, anti-oxidant, anti-bacterial, cell proliferation, and wound healing effects [31,48,49,50,51]. Our previous study suggested that SP not only reduces lipid accumulation, but also induces fatty acid oxidation and browning in adipose tissue, so may represent a potential therapeutic material for obesity and obesity-related metabolic complications [33]. According this study, the nutrient composition of the SP is 89.80% protein, 6.78% carbohydrate, 1.79% sodium, 0.94% sugar, and 0.01% fat. Additionally, the SP used in this study contains small dipeptides and tripeptides, so may help with efficient intestinal absorption [52]. Furthermore, the major free amino acid composition of SP was 33.10% glycine, 28.1% alanine, and 11.1% serine on a dry-matter basis. However, the effects of SP on glucose tolerance and sarcopenic obesity had not yet been investigated. Therefore, we determined the effects of SP on metabolism and muscle differentiation using HFD-fed mice, and 3T3-L1 and C2C12 cell lines.

Obesity is a well-recognized risk factor for T2D, which develops as hyperglycemia and glucose intolerance worsen. In addition, in recent years, the deleterious effects of abdominal obesity and loss of skeletal muscle mass on metabolism in older people have become apparent. With aging, the WAT distribution shifts from predominantly SAT to VAT. The ensuing abdominal obesity advances the onset of age-associated diseases, such as T2D, by inducing insulin resistance [53]. In particular, VAT has a significant role in the control of insulin sensitivity [54]. In addition, because skeletal muscle is the principal tissue for glucose uptake and utilization, a loss of muscle mass increases glucose intolerance, thereby further increasing the risk of T2D. However, muscle loss results in physical dysfunction, as well as metabolic disease [55]. According to a recent study, a 10% increase in the ratio of muscle to total body mass results in an 11% reduction in the risk of insulin resistance [56]. Indeed, a recent study reported that SP modulates the glucose level and insulin recreation in a partial pancreatectomized rat model, so may prevent non-obese T2D [57]. The aim of the present study was to investigate the effect of SP on obesity-related hyperglycemia, the dysregulation of muscle metabolism, and the decline in physical function.

We found that SP inhibited weight gain, probably by reducing the differentiation of pre-adipocytes, secondary to a lower expression of C/EBPα and PPARγ, which was identified in both VAT and 3T3-L1 adipocytes. These transcription factors play a role in the storage of energy in the form of triglyceride. SP administration increased the rectal temperature of mice, which probably implies that the body mass loss was due to an increase of energy expenditure, not due to a lower food intake. To assess whether SP affects body temperature related to infection in mice, serum creatinine levels, AST, and ALT were measured. Creatinine is a biomarker used to determine renal function, which is primarily produced in muscle [58]. Additionally, liver function parameters, including AST and ALT, are representative factors of lipid damage. However, there was no significant difference in these serum concentrations between the mice treated with SP and other groups. Consequently, our data indicated that the effect of SP on kidney or liver injury may preclude. Furthermore, our previous study reported that SP induces lipolysis and fatty acid oxidation by increasing the expression level of phosphorylated hormone-sensitive lipase (p-HSL), peroxisome proliferator-activated receptor alpha (PPARα), and carnitine palmitoyltransferase 1 (CPT1) in VAT, so SP may reduce the amount of fat in adipocytes [33]. Furthermore, it has been reported that SP induces WAT-to-BAT trans-differentiation by upregulating (Uncoupling protein 1) UCP1, so SP may dissipate fat to heat in adipocytes. In addition, adiponectin expression was higher in the VAT of SP-treated groups than in HFD-fed obese mice. An increase in adiponectin expression in VAT would be expected to result in higher secretion, and greater insulin-sensitizing and anti-diabetic effects. SP also ameliorated the hyperglycemia in HFD-fed mice and reduced the circulating concentrations of total cholesterol and triglyceride. Therefore, our findings suggest that dietary SP inhibits lipid accumulation by reducing adipogenesis and regulating lipid metabolism, which implies that SP may be capable of reducing obesity and ameliorating obesity-associated metabolic disease.

SP, when administered at 50 or 200 mg/kg/day, also prevented the HFD-induced increase in fasting glucose and ameliorated glucose intolerance in HFD-fed mice. In addition, the phosphorylation of insulin-signaling intermediates and the expression of GLUT4 were increased in the VAT and skeletal muscle of HFD-induced obese mice treated with SP for 6 weeks. IRS and AKT are key intermediates in the insulin signaling pathway and mediate insulin-stimulated glucose disposal in both adipocytes and muscle cells [59]. The most important factor in the maintenance of blood glucose is glucose transport, which is principally performed by GLUT4 expressed in skeletal muscle after a meal [60]. The present data also suggest that SP enhances muscle energy expenditure by upregulating transcription factors involved in mitochondrial biogenesis: PGC-1α NRF1 and UCP3, which themselves upregulate GLUT4 expression [61]. Recent research reported that muscle mitochondrial-related genes, including NRF1, PGC1α, and UCP3, regulate insulin resistance during obesity [62,63]. In particular, PGC1α is a well-known transcription factor correlated with sarcopenia and metabolic disease during aging [64,65]. Moreover, UCP3 is known to be primarily found in skeletal muscle and has an important role in modulating mitochondrial biogenesis and energy expenditure in skeletal muscle via AMPK activation [28]. Indeed, the expression of these molecules was increased in skeletal muscle from HFD-fed mice and C2C12 cells treated with SP. Therefore, our finding indicates that SP increased the expression level of these proteins, implying that SP may increase the mitochondrial respiration in skeletal muscle in HFD-induced obese mice. Furthermore, we have shown that SP increases AMPK activation in muscle tissue and C2C12 myoblasts, which would also be expected to increase glucose uptake by inducing the translocation of GLUT4 to the plasma membrane of adipose tissue or skeletal muscle, independently of insulin, thereby helping to limit hyperglycemia [66]. The effects of SP on gene expression in C2C12 cells was compared with those of Met, which is an anti-T2D drug that ameliorates hyperglycemia, at least in part by increasing AMPK phosphorylation [67]. Therefore, SP has potential for use in the treatment of T2D and obesity.

According to recent research, obesity also leads to the progressive loss of skeletal muscle mass and strength, which can cause further health problems [68]. The obesity-related loss of skeletal muscle, referred to as sarcopenia, is associated with the dysregulation of glucose metabolism and accelerated loss of muscle loss [69]. Therefore, a number of studies have focused on the mechanisms of this skeletal muscle dysfunction, as well as interrogating potential nutritional interventions, including the use of diets rich in protein, which could help retain muscle strength [70,71]. The present study has shown that SP increases the expression of genes involved in skeletal muscle differentiation in mice. Specifically, HFD-induced obesity was associated with inhibition of the MyoD/Myogenin pathway in mice skeletal muscle. However, the administration of SP increased the expression of these proteins in skeletal muscle. Recent studies have reported that myogenesis is regulated by myogenic regulatory factors such as MyoD, Myogenin, and MYHs [72]. MyoD is dispensable for skeletal muscle development and regulates the skeletal myogenic developmental program, which induces myogenic markers such as MYHs and Myogenin [73]. Additionally, Myogenin is required for myoblast fusion into myotubes. Our results suggest that SP treatment increases myoblast differentiation by myogenic factors such as MyoD, Myogenin, and MYH3 in C2C12, so SP may thus improve muscle development. Consistent with this, the present study has shown that the HFD-induced reduction in grip strength per unit body mass was ameliorated by SP. Indeed, HFD induced typical features of muscle wasting, such as lower fiber CSA values. Lastly, our last in vivo data showed that the age-related loss of muscle mass and strength can be restored by the impaired recovery of muscle mass following the treatment of SP in 12-month-old mice. In particular, SP increased hindlimb muscle so that the physical activity of the adult mice was increased. While the body mass of AM + SP250 displays no significant difference by SP treatment, it was slightly decreased compared to AM. Collectively, SP treatment inhibited the age-related reduction in muscle mass in adult mice; thereby, the muscle strength and physical function were recovered. Therefore, SP supplementation is considered to enhance physical strength, as well as improve an abnormal status, among the elderly. Considering this, SP may be useful as a food supplement for limiting the development of sarcopenia alongside obesity or aged-induced obesity.

To overcome sarcopenia, it is important to not only stimulate myogenesis, but to also inhibit muscle wasting. Despite the growing amount of research on the treatment of sarcopenia, the molecular mechanisms that control obesity-related change in muscle mass are not fully understood. The E3-ligase F-box protein system is one of the major pathways that modulate muscle degradation, and it plays a vital role in regulating muscle mass. Specifically, activation of the ubiquitin protease pathway in muscle atrophy is associated with increases in the expression of two muscle-specific proteins: Fbx32 and MuRF1 [74]. Furthermore, HFD consumption causes skeletal muscle to overexpress these proteins. In the present study, SP treatment reduced the expression level of those involved in muscle degradation in HFD-fed obese mice. We found that the expression of Fbx32 and MuRF1 was lower in the muscle of SP-treated groups than in the HFD-fed group. Therefore, our results indicate that SP may have the potential to prevent or reverse muscle atrophy by modulating the expression of these ubiquitine proteases.

In conclusion, the present study has shown that SP inhibits weight gain, reduces fat accumulation, and increases glucose uptake in the VAT of HFD-fed obese mice. In addition, we have provided evidence that SP promotes myoblast differentiation and mitochondrial biogenesis, and reduces the expression of ubiquitin ligases in skeletal muscle. Therefore, SP might prove to be a beneficial dietary supplement for the prevention of obesity and hyperglycemia in association with sarcopenia.

## Figures and Tables

**Figure 1 cells-09-00377-f001:**
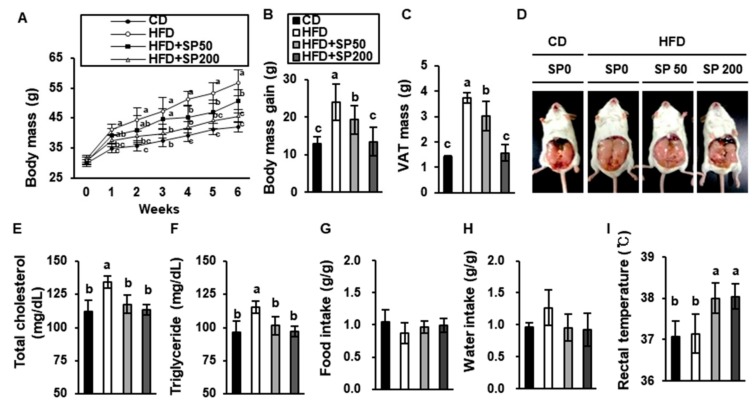
Effect of silk peptide consumption on the development of high-fat diet-induced obesity in mice. (**A**) Weekly body mass measurements and (**B**) body mass gain of mice treated for 6 weeks. (**C**) Representative photographs of the mice. (**D**) Effect of silk peptide (SP) on visceral adipose tissue (VAT) mass in chow diet (CD)- and high-fat diet (HFD)-fed groups. Serum (**F**) total cholesterol and (**E**) triglyceride concentrations were measured using colorimetric kits. (**G**) Food intake and (**H**) water consumption per unit body mass. Data are expressed as the mean ± SD (*n* = 8). Values with different letters are significantly different; *p* < 0.05 (a > b > c).

**Figure 2 cells-09-00377-f002:**
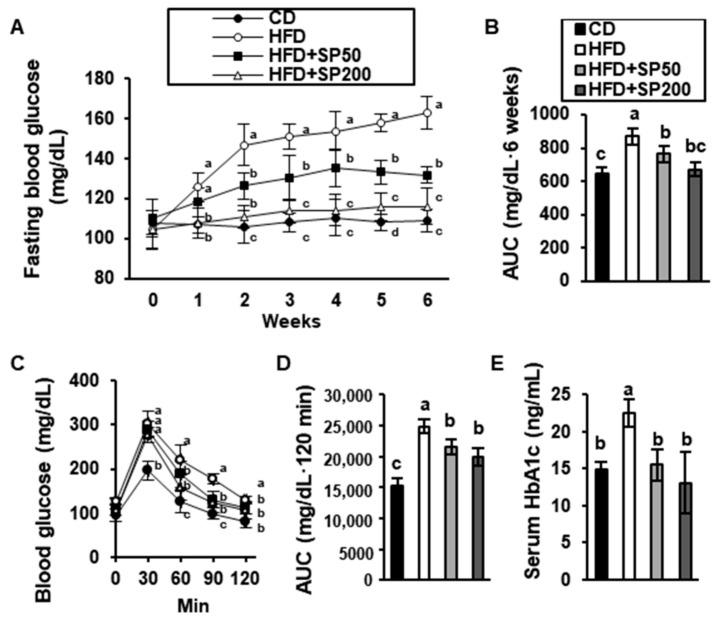
Effect of silk peptide on blood glucose in high-fat diet-fed mice. (**A**) Fasting blood glucose was measured and (**B**) the area under the curve (AUC) over time was calculated. (**C**) Oral glucose tolerance testing (OGTT) was performed after 6 weeks and (**D**) its AUC was calculated. (**E**) HbA1c was measured using a commercial kit. Data were analyzed using one-way ANOVA and Duncan’s test. Values with different letters are significantly different; *p* < 0.05 (a > b > c > d).

**Figure 3 cells-09-00377-f003:**
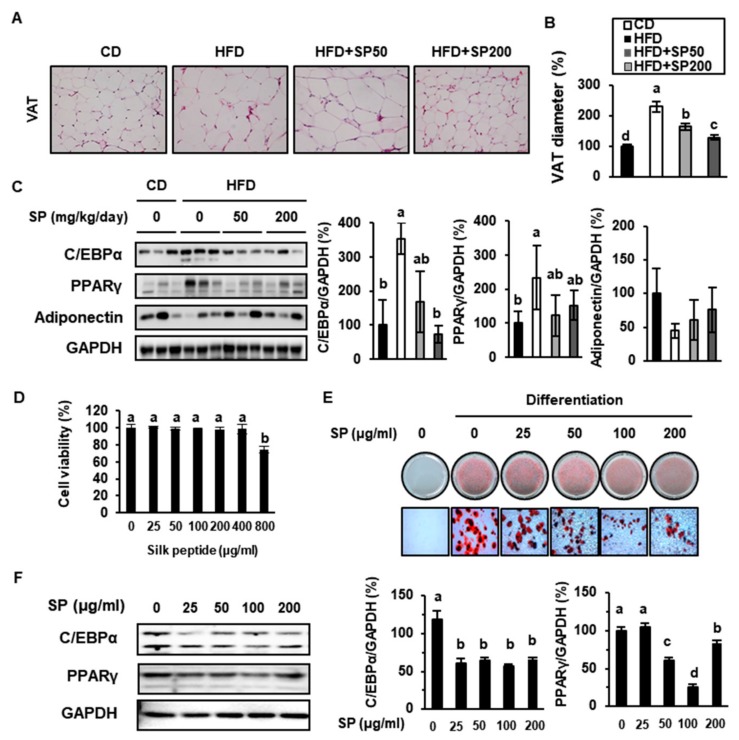
Effect of silk peptide on lipid accumulation in visceral adipose tissue from high-fat diet-fed mice and 3T3-L1 cells. (**A**) Hematoxylin and eosin staining of VAT from mice treated for 6 weeks. (**B**) Mean diameter of VAT cells (arbitrary units). (**C**) The expression of adipogenic factors in VAT was measured by western blotting. (**D**) Viability of 3T3-L1 pre-adipocytes treated with SP for 24 h. (**E**) Oil red O was used to stain cells after 10 days of differentiation in the presence or absence of SP, and photomicrographs were obtained (400× magnification). (**F**) Expression levels of proteins involved in adipogenesis in 3T3-L1s. Data are the mean ± SD of six replicates and were analyzed using one-way ANOVA and Duncan’s test. Values with different letters are significantly different; *p* < 0.05 (a > b > c > d).

**Figure 4 cells-09-00377-f004:**
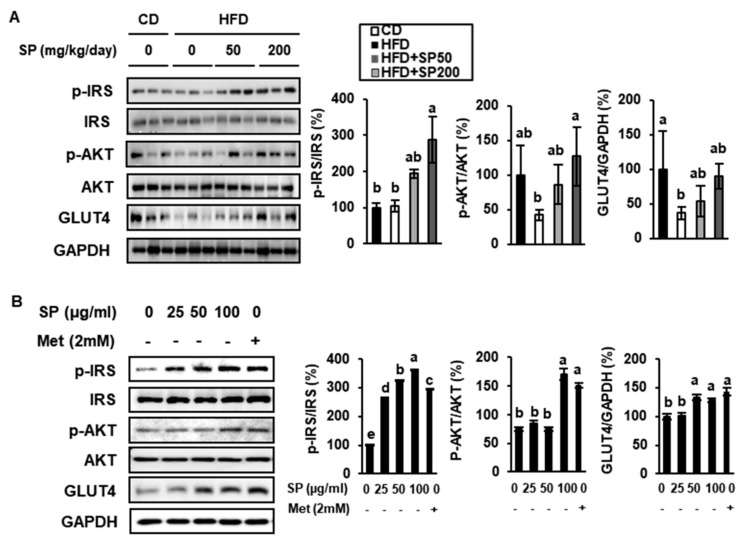
Effect of silk peptide on insulin resistance in visceral adipose tissue from high-Figure 3. T3- L1 cells. (**A**) Expression of proteins involved in glucose disposal (p-IRS, IRS, p-AKT, AKT, and GLUT4) in VAT, analyzed by western blotting. (**B**) Expression levels of proteins involved in insulin sensitivity in 3T3-L1s compared with cells treated with 2 mM metformin (Met). The data were analyzed using one-way ANOVA and Duncan’s test. Values with different letters are significantly different; *p* < 0.05 (a > b > c > d > e).

**Figure 5 cells-09-00377-f005:**
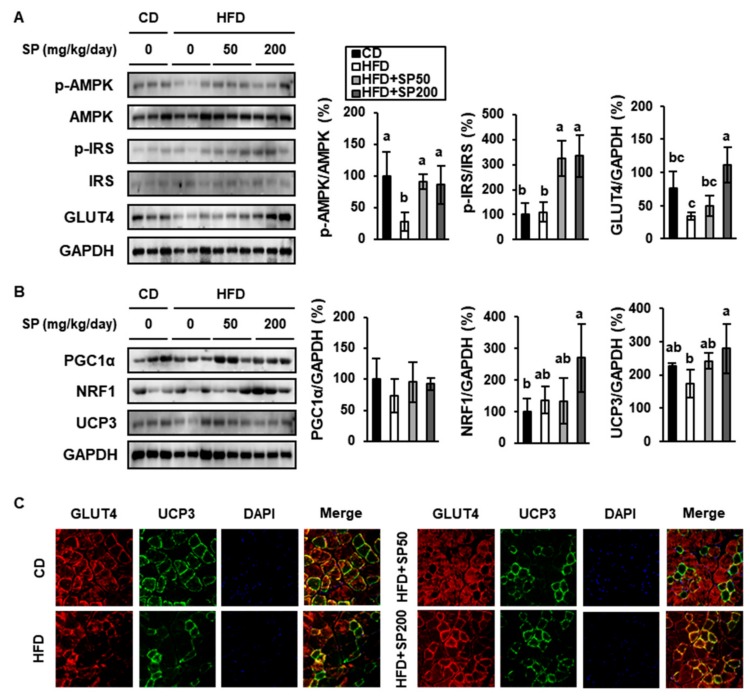
Effect of silk peptide on proteins involved in glucose uptake and mitochondrial metabolism in muscle from high-fat diet-fed mice. (**A**) Expression of proteins involved in glucose uptake (p-AMPK, AMPK, p-IRS, IRS, and GLUT4) was determined by western blotting. (**B**) Expression of proteins involved in mitochondrial biogenesis (PGC1α, NRF1, and UCP3) was determined by western blotting. (**C**) Immunofluorescence images of muscles were captured at 400× magnification. Muscles were fixed with methanol, and then anti-GLUT4 or anti-UCP3 antibodies and DAPI were applied. Data were analyzed using one-way ANOVA and Duncan’s test. Values with different letters are significantly different; *p* < 0.05 (a > b > c).

**Figure 6 cells-09-00377-f006:**
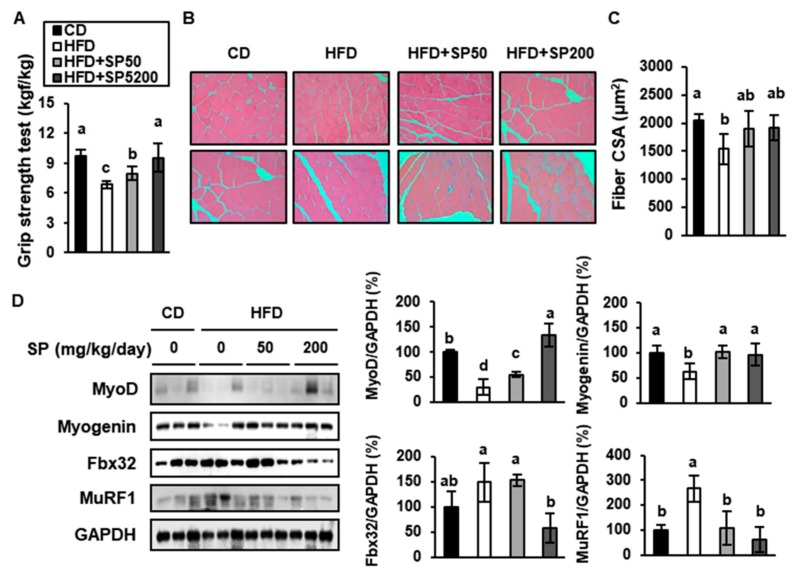
Effect of silk peptide on proteins mediating muscle differentiation and sarcopenia in HFD-fed mice. (**A**) Grip strength was measured after 6 weeks of treatment. (**B**) Hematoxylin and eosin staining of muscle from mice treated for 6 weeks (*n* = 2). (**C**) Quantification of the cross-sectional area (CSA) of muscle fibers. The CSA of each muscle fiber in each field was measured using the Image J program. (**D**) Expression levels of proteins involved in muscle differentiation (MyoD and Myogenin) and sarcopenia (Fbx32, MuRF1) were determined using western blotting. Data were analyzed using one-way ANOVA and Duncan’s test. Values with different letters are significantly different; *p* < 0.05 (a > b > c > d).

**Figure 7 cells-09-00377-f007:**
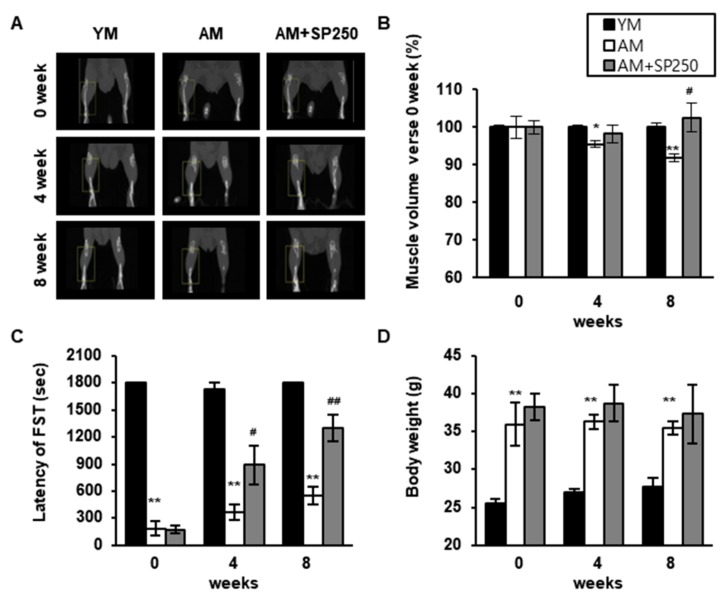
Effect of SP on muscle size and strength, and body weight in adult mice. (**A**) Micro-CT image of hindlimb muscle. (**B**) Relative hindlimb muscle volumes were quantitated compared with those of 0 week. (**C**) Latency of the forced swimming test was measured for 30 min. (**D**) Body weight of the mice after 8 weeks of the experimental period. ^*^
*p* < 0.05, ^**^
*p* < 0.01 compared to young mice group at 0 week; ^#^
*p* < 0.05, ^##^
*p* < 0.01 compared to adult mice group at 0 week.

**Figure 8 cells-09-00377-f008:**
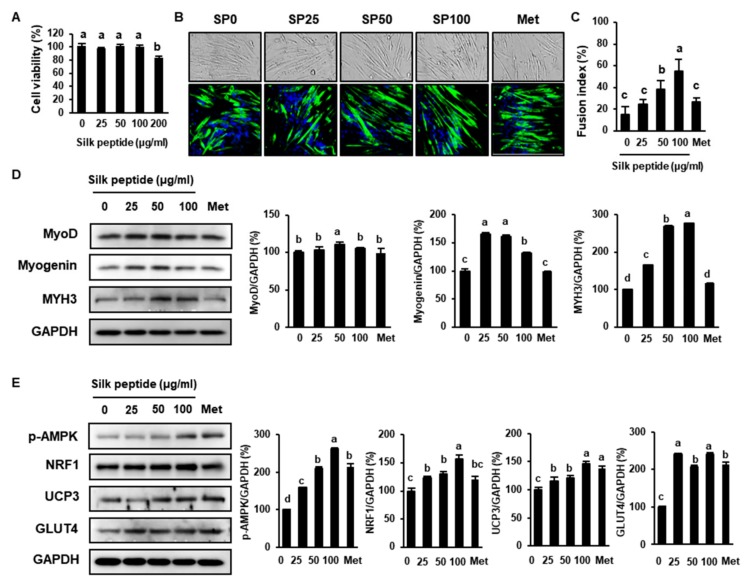
Effect of silk peptide on myoblast differentiation and glucose uptake in C2C12 cells. (**A**) Viability of C2C12 cells treated with SP for 24 h. (**B**) Images of morphological change of C2C12 were obtained by an optical microscope (up, 200×). Immunofluorescence for the MYH3 expression of C2C12 in the presence or absence of SP (down, 400×). (**C**) Fusion index was calculated at the end of differentiation by dividing the number of nuclei within multinucleated myofibers by the total number of nuclei. (**D**) Expression levels of proteins involved in muscle differentiation (MyoD, Myogenin, and MYH3) were determined using western blotting. (**E**) Expression levels of mitochondrial proteins and proteins involved in glucose uptake (p-AMPK, PGC1α, UCP3, and GLUT4) analyzed by western blotting. The data were analyzed using one-way ANOVA and Duncan’s test. Values with different letters are significantly different; *p* < 0.05 (a > b > c > d). Met; 2 mM metformin.

**Table 1 cells-09-00377-t001:** Effect of SP treatment on organ weight in HFD-fed mice for 6 weeks.

Organs	Organ Weight (g)			
	CD	HFD	HFD + SP50 *	HFD + SP200 ^*^
Visceral white adipose tissue (VAT)	1.42 ± 0.05 ^c^	3.74 ± 0.51 ^a^	3.03 ± 0.57 ^b^	1.57 ± 0.33 ^c^
Subcutaneous white adipose tissue (SAT)	0.74 ± 0.18 ^c^	1.67 ± 0.51 ^a^	1.43 ± 0.44 ^b^	0.95 ± 0.28 ^c^
Brown adipose tissue (BAT)	0.13 ± 0.03 ^a^	0.13 ± 0.0.3 ^a^	0.12 ± 0.04 ^a^	0.11 ± 0.02 ^a^
Liver	1.59 ± 0.21 ^a^	1.69 ± 0.15 ^a^	1.62 ± 0.32 ^a^	1.46 ± 0.14 ^a^
Lung	0.23 ± 0.03 ^a^	0.24 ± 0.02 ^a^	0.22 ± 0.02 ^a^	0.24 ± 0.02 ^a^
Kidney	0.64 ± 0.08 ^a^	0.67 ± 0.07 ^a^	0.63 ± 0.08 ^a^	0.63 ± 0.04 ^a^
Spleen	0.13 ± 0.02 ^a^	0.13 ± 0.02 ^a^	0.13 ± 0.02 ^a^	0.13 ± 0.06 ^a^

* (mg/kg/day), data are expressed as the mean ± SD (*n* = 6). Values with different letters are significantly different; *p* < 0.05 (a > b > c).

**Table 2 cells-09-00377-t002:** Effect of SP treatment on the blood parameter in HFD-fed mice.

Group	Blood Parameter (mg/dL)			
	CD	HFD	HFD + SP50 ^*^	HFD + SP200 ^*^
Creatinine	0.22 ± 0.04 ^a^	0.25 ± 0.04 ^a^	0.22 ± 0.03 ^a^	0.23 ± 0.02 ^a^
Aspartate aminotransferase (AST)	36.67 ± 7.37 ^a^	41.33 ± 5.51 ^a^	35.53 ± 9.07 ^a^	35.33 ± 3.51 ^a^
Alanine aminotransferase (ALT)	70.33 ± 9.50 ^a^	76.00 ± 5.20 ^a^	64.67 ± 12.42 ^a^	62.67 ± 8.33 ^a^

^*^ (mg/kg/day), data are expressed as the mean ± SD (*n* = 6). Values with different letters are significantly different; *p* < 0.05 (a > b).

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
