# Peer review of "Effect of Dietary Silk Peptide on Obesity, Hyperglycemia, and Skeletal Muscle Regeneration in High-Fat Diet-Fed Mice"

_cells, 2020, doi:10.3390/cells9020377_

Round 1

Reviewer 1 Report

The authors investigated the effects of silk peptide on high-fat diet (HFD)-induced obesity and found that, in HFD-fed mice, silk peptide inhibited body mass gain and visceral adipose tissue mass, and increased glucose uptake. Moreover, silk peptide promoted myogenesis and mitochondrial biogenesis. The manuscript shows that silk peptide may be useful for the treatment of obesity-associated hyperglycemia and sarcopenia. This work is very carefully carried out and, overall, the data are convincing.

Minor comments:

In Figure 3, the authors wrote “effect of silk peptide on lipi”. The sentence should be correct.

The authors detected the expression levels of p-IRS, p-AKT and p-AMPK. The authors should indicate these phosphorylation site.

In Figure 6, the authors performed grip strength test. The method should be described.

In Figure 7, is “p-MAPK” “p-AMPK”?

Author Response

Dear Editor and reviewers,

Thank you for considering our manuscript for publication in Cells. We are very pleasure to have been given the opportunity to revise our manuscript, “Effect of Dietary Silk Peptide on Obesity, Hyperglycemia, and Skeletal Muscle Differentiation in High-fat Diet-Fed Mice”. We have addressed the reviewer’s comments point-by-point and made the necessary changes to the manuscript.

We hope that the manuscript is acceptable for publication in Cells. We appreciate to consider this paper for publication in Cells, and declare that authors of this work have no conflict of interests.  

Sincerely yours,

Boo-Yong Lee, Ph.D.

Revision of Cells-702946 (Please see the attachment)

1-In Figure 3, the authors wrote “effect of silk peptide on lipi”. The sentence should be correct.

Answer) Thank you for your pointing out our mistake. As your kind advice, we revised as “Figure 3. Effect of silk peptide on lipid accumulation in visceral adipose tissue from high-fat diet-fed mice and 3T3-L1 cells”.

2-The authors detected the expression levels of p-IRS, p-AKT and p-AMPK. The authors should indicate these phosphorylation site.

Answer) As reviewer’s comment, we added phosphorylation site of each factor presented in “2.1 Materials” on page 3 line 100-102, as “Anti-C/EBPα, PPARγ, myogenin, AKT, p-AKT (Ser 473), AMPK, and p-AMPK (Thr 172) antibodies were obtained from Cell Signaling Technology (Danvers, MS, USA). Anti-DGAT1, GAPDH, Fbx32, IRS, and p-IRS (Thr 632) antibodies were purchased from Santa Cruz Biotechnology, CA, USA.”.

3-In Figure 6, the authors performed grip strength test. The method should be described.

Answer) As reviewer’s comment, we added the 2.6 Grip strength test on Page 3; “2.6 Grip strength test: The grip strength of the mice was assessed using a grip strength meter with single sensor, which called Chatillon force measurement system (Columbus Instrument, OH, USA). The mice were placed with their forelimbs and hindlimb on a narrow bar and the strength when mice fall was measured at the end of the oral administration period.”

4-In Figure 7, is “p-MAPK” “p-AMPK”?

Answer) We changed the “p-MAPK” to “p-AMPK” as you comment. Also, we changed “Myogenine” to “Myogenin” Figure 6-7 and revised “MuRF” to “MuRF1” in Figure 7. Thank you for your pointing out our mistake.

Reviewer 2 Report

In this study the authors have investigated the role of Silk peptide on obesity, hyperglycemia and skeletal muscle in high fat diet-fed mice. They reported that Silk peptide inhibits body mass gain, improves glucose tolerance and induces change at skeletal muscle level.

This paper is interesting although the conclusions are overstated. I have some major criticisms.

Major criticisms:

1-Your mice on HF diet had a weight gain of 25g in 6 weeks which is considerable. How many animals per cage? At what temperature the animals housed?

2-In what nutritional state the mice were euthanized? In fasting or feeding condition? This is very important for the subsequent analyses of insulin signaling pathway performed on VAT and on skeletal muscle.

3-The authors found SP administration induced a resistance to obesity on HF diet but they indicated that the inhibition of body mass gain in SP-treated mice was not due to lower food intake. In this case, the weight loss is due either to a sharp increase in basal metabolism or a drop in intestinal absorption or a combination of both.

In order to answer this central question, metabolic analyses of these mice are necessary, for example using the Comprehensive Lab Animal Monitoring System (CLAMS; Columbus Instruments). Have you studied body temperature in your mice?

4-The authors observed that fasting blood glucose was lower in SP treated mice and that these mice had an increase glucose tolerance compared to HF diet group.

These results suggest better insulin sensitivity or an increase insulin secretion. What about insulin levels? What about insulin sensitivity? Have you performed ITT?   

5-To determine whether SP administration had an anti-obesity effect the author analyzed adipose tissue.

What about fat mass? VAT weight?

6-The authors claim that SP increases glucose absorption in fat tissue only on the basis of western-blot experiments.

Have you done glucose uptake experiments?

7-If SP increases the absorption of glucose into the adipose tissue then you should have an increase in the amount of lipids in adipocytes. However, ORO staining in 3T3-L1 indicates that lipid accumulation in these cells was substantially prevented by SP.

How do you reconcile these two results?  

7-Depending on the skeletal muscle studied, the contractile and metabolic characteristics are very different.

Which muscles did you use for the Western-block and immuno-fluorescence experiments?

8-Skeletal muscle is characterized by its mass, contractile type and metabolic activity.

What is the mass of muscles such as the quadriceps, gastrocnemius, tibialis?

Are there changes in the expression of myosin heavy chains?

Are there changes in the activity of mitochondrial respiratory chain complexes?

9-In figure 6, the authors claim that SP treatment increased fiber cross-section area.

If the authors have made analyses of the whole section area and quantified the number and surface area of the fibers it would be nice to show this.

10-Furthermore, the authors claim that SP treatment increased MyHC expression in HF diet. This statement is nonsense. The muscle remains muscle and it is mostly composed of MyHC. Their antibody is probably directed against a type of myosin and their western-blots suggest rather that SP induces a change in the contractile type.

Which isoform of MyHC recognizes your antibody?

11-It is essential that you indicate the exact references of the antibodies used. For example, some phosphorylations activate AKT while others inhibit it. The interpretation is not the same...

12-The authors claim that SP may prevent sarcopenia by enhancing muscle differentiation and inhibiting the muscle atrophy induced by HF diet.

Influence of SP on muscle Weight? Have you studied autophagy or synthesis flux? Have you studied the influence of SP on skeletal muscle regeneration?

13-The authors claim that SP stimulates myoblast differentiation.

Did you measure the fusion index to state this?

In conclusion, the data reported in this work are interesting and clearly indicated that SP have anti-obesity effect but this work must be completed because a lot of data are missing and made interpretation hazardous. The discussion will have to be reworked in the light of the replies received to the questions asked.

Author Response

Dear Editor and reviewers,

Thank you for considering our manuscript for publication in Cells. We are very pleasure to have been given the opportunity to revise our manuscript, “Effect of Dietary Silk Peptide on Obesity, Hyperglycemia, and Skeletal Muscle Differentiation in High-fat Diet-Fed Mice”. We have addressed the reviewer’s comments point-by-point and made the necessary changes to the manuscript.

In major revision, we have addressed the reviewer’s comment about method of grip strength test, experiment to investigae the body temperature, and explanation of muscle data implying that SP has a potential to treatment for sarcopenia using 6-week-old mice and 12-month-old mice. Therefore, we revised regarding data and added supplementary data and related description in result and discussion section, and highlighted the changes with track changes in the manuscript.

We hope that the manuscript is acceptable for publication in Cells. We appreciate to consider this paper for publication in Cells, and declare that authors of this work have no conflict of interests.  

Sincerely yours,

Boo-Yong Lee, Ph.D.

Revision of Cells-702946 (Please see the attachment)

Reviewer2

1-Your mice on HF diet had a weight gain of 25g in 6 weeks which is considerable. How many animals per cage? At what temperature the animals housed?

 Answer) We appreciate your suggestion. As reviewer’s comment, we revised as “The mice were initially housed for 1 week under a 12 h light/dark cycle condition in 20-24 ℃ temperature and 44-52% humidity to permit adaptation. Four male mice were kept in each cage used in the study. After adaptation period, the mice …” in 2.2 Animals and experimental design on page 3.

2-In what nutritional state the mice were euthanized? In fasting or feeding condition? This is very important for the subsequent analyses of insulin signaling pathway performed on VAT and on skeletal muscle.

 Answer) As reviewer’s comment, we corrected in as “At the end of the experimental period, the mice were fasted for 12 h and euthanized using gradual-fill method of carbon dioxide euthanasia, and their tissues were collected for analysis.” on page 3

3-The authors found SP administration induced a resistance to obesity on HF diet but they indicated that the inhibition of body mass gain in SP-treated mice was not due to lower food intake. In this case, the weight loss is due either to a sharp increase in basal metabolism or a drop in intestinal absorption or a combination of both.

In order to answer this central question, metabolic analyses of these mice are necessary, for example using the Comprehensive Lab Animal Monitoring System (CLAMS; Columbus Instruments). Have you studied body temperature in your mice?

 Answer) Thank you for your suggestion. We has been tried to investigate the lipid metabolism as your advice. However, we could not analyze the metabolic change using the CLAMS due to financial problem. Thereby, we measured the body temperature of the mice to investigate the effect of SP on energy expenditure, instead of CLAMS. We added this data as Figure 1I, and explained the result:

“Lastly, to investigate the effect of SP on energy expenditure, rectal temperature of the mice was measured at the end of the administration period. As a result, SP significantly increased body temperature to 38.0 ± 0.4 and 38.0 ± 0.3 ℃ at 50 and 200 of SP treated group, respectively.”-in Result 3.1.

“Moreover, SP administration increased the rectal temperature of the mice, which probably implies the body mass loss was due to an increase of energy expenditure not due to lower food intake.”-Discussion.

Moreover, in this regard, our previous study determined that browning effect of SP on white adipose tissue in HFD-induced obese mice model . Please refer to this article [1]; Dietary Silk Peptide Prevents High-Fat Diet-Induced Obesity and Promotes Adipose Browning by Activating AMP-Activated Protein Kinase in Mice, Nutrients, 12(1), 2020, DOI: 10.3390/nu12010201.

4-The authors observed that fasting blood glucose was lower in SP treated mice and that these mice had an increase glucose tolerance compared to HF diet group.

These results suggest better insulin sensitivity or an increase insulin secretion. What about insulin levels? What about insulin sensitivity? Have you performed ITT?   

 Answer) Thank you for your suggestion. As your comment, our findings showed that SP may regulate insulin sensitivity or insulin secretion in HFD-induced obese mice. Even though we tried to ITT at the end of the experimental period, we could not analyze ITT due to technical problem, unfortunately. However, we suggest an article in Nutrients (in processing) to explain glucose homeostasis of SP in partial pancreatectomized rat model [2]. In this study, they also used same sample of SP obtained from Worldway Co., Ltd. and determined glucose level by fasting or intraperitoneal insulin tolerance test (IPITT) as well as insulin secreation. As a result, SP significantly reduced serum glucose concentrations in fasted and post-prandial states in Px rats to levels similar to those observed compared with control group. Also, their study indicates that SP dose-dependently not only reduced serum glucose but also elevated glucose-stimulated insulin secretion. Therefore, we added “The recent study reported that SP modulates the glucose level and insulin recreation in partial pancreatectomized rat moel, thereby SP may prevent non-obese T2D.“ in discussion section.

5-To determine whether SP administration had an anti-obesity effect the author analyzed adipose tissue.

What about fat mass? VAT weight?

 Answer) As shown in Figure 1C, we presented the VAT weight of the mice treated with SP was significantly decreased. The reason why we measured VAT mass is to investigate whether VAT is especially associated with anti-obesity effect and metabolic disease correlated glucose uptake capacity. However, we also observed the SAT, BAT, liver, lung, kidney, and spleen mass of the mice, so therefore added Table 1 to better understanding. Also, we revised Result 3.1 as exact explanation.

Table 1. Effect of SP treatment on organ weight in HFD-fed mice for 6 weeks.

Organs

Organ weight (g)

CD

HFD

HFD+SP50 *

HFD+SP200 *

Visceral white adipose tissue (VAT)

1.42 ± 0.05 c

3.74 ± 0.51 a

3.03 ± 0.57 b

1.57 ± 0.33 c

Subcutaneous white adipose tissue (SAT)

0.74 ± 0.18 c

1.67 ± 0.51 a

1.43 ± 0.44 b

0.95 ± 0.28 c

Brown adipose tissue (BAT)

0.13 ± 0.03 a

0.13 ± 0.0.3 a

0.12 ± 0.04 a

0.11 ± 0.02 a

Liver

1.59 ± 0.21 a

1.69 ± 0.15 a

1.62 ± 0.32 a

1.46 ± 0.14 a

Lung

0.23 ± 0.03 a

0.24 ± 0.02 a

0.22 ± 0.02 a

0.24 ± 0.02 a

Kidney

0.64 ± 0.08 a

0.67 ± 0.07 a

0.63 ± 0.08 a

0.63 ± 0.04 a

Spleen

0.13 ± 0.02 a

0.13 ± 0.02 a

0.13 ± 0.02 a

0.13 ± 0.06 a

* (mg/kg/day), Data are expressed as mean ± SD (n = 6). Values with different letters are significantly different, p < 0.05 (a > b > c).

6-The authors claim that SP increases glucose absorption in fat tissue only on the basis of western-blot experiments.

Have you done glucose uptake experiments?

 Answer) Thank you for your suggestion, but we could not assess glucose uptake by using the ELISA kit using 2-deoxyglucose due to the deficiency blood serum volume from the mice. However, our western data of VAT and skeletal muscle indicates that SP effectively increased the expression level of p-AMPK, p-AKT, p-IRS, and GLUT4. In addition we also estimated the effect of SP on HFD-induced hyperglycemia by oral glucose tolerance test (OGTT) and hemoglobin A1c (HbA1c) assay kit. GTT, known as the oral glucose tolerance test, is widely used to assess how body is able to absorb glucose after consuming a specific amount of sugar. Also, HbA1c test provides information about average of blood glucose level over the past 2-3 month. Therefore both of tests were used to check for diabetes or prediabetes in our mice model. Taken together, we suggest that SP has a potent to regulate glucose uptake in HFD-induced obese mice. Moreover, we suggest a patent regarding the ant-diabetes effect of SP from Worldway Co., Ltd. In this patent, it is explored that SP regulates glucose level in C/57db/db mice model. Please refer to this patent-“A composition comprising peptide for treatment or prevention of diabetes mellitus, Publication number and date: 1020100020145 (05.03.2010)”.

7-If SP increases the absorption of glucose into the adipose tissue then you should have an increase in the amount of lipids in adipocytes. However, ORO staining in 3T3-L1 indicates that lipid accumulation in these cells was substantially prevented by SP.

How do you reconcile these two results?  

 Answer) As your advice, adipose tissue not only regulates lipid balance but also serves as a crucial integrator of glucose homeostasis. Also, lipid can be accumulated in early or middle differentiation of adipocytes, when glucose absorption was increased in adipocytes, as your advice. For instance, metformin is a widely used drug in the therapy of patients affected by diabetes mellitus and is a medication that many these people take to control blood glucose level by improved insulin sensitivity. However, metformin can induce fat accumulation again as an its side effect. In this regard, SP has a potent to regulate fasting glucose level without this side effect. In this study, we evaluated glucose uptake via GLUT4 expression in the mature differentiated 3T3-L1 for 8 days, thereby ORO staining data indicates the lipid accumulation after late-differentiated of adipocytes. In addition, we previously investigated that SP induced lipolysis and fatty acid oxidation by increasing the expression level of phosphorylated hormone-sensitive lipase (p-HSL), peroxisome proliferator-activated receptor alpha (PPARα), and carnitine palmitoyltransferase 1 (CPT1) in VAT. Furthermore, we checked that SP induces WAT-to-BAT trans-differentiation by upregulating (Uncoupling protein 1) UCP1, thereby SP may dissipate fat to heat in adipocytes [1]. We revised this explain in discussion section.

8-Depending on the skeletal muscle studied, the contractile and metabolic characteristics are very different.

Which muscles did you use for the Western-block and immuno-fluorescence experiments?

 Answer) We used gastrocnemius muscle of mice for skeletal muscle analysis in this study. Gastrocnemius muscle is one of the large muscles of the leg, and this muscle is associated with out grip strength test data. Regarding on it, we have conducted experiments such as western blot and immusostaining analysis with reference to many papers, reported the effect of various dietary materials on gastrocnemius muscle in obese mice [3, 4]. Therefore, we correctly wrote the name of the muscle in manuscript.

9-Skeletal muscle is characterized by its mass, contractile type and metabolic activity.

What is the mass of muscles such as the quadriceps, gastrocnemius, tibialis?

Are there changes in the expression of myosin heavy chains?

Are there changes in the activity of mitochondrial respiratory chain complexes?

 Answer) To study skeletal muscle physiology, we wanted to isolate the muscle separate the gastrocnemius, tibialis anterior, and soleus muscle from the HFD-fed obese mice. Since it was our first time to research the muscle or sarcopenic obesity, so therefore only gastrocneminus muscle can be analyzed with the help of other researchers. We also feel sorry for this situation and please understand it.

Instead, we suggest some data added as a supplementary data, which is observed before this present study, as your kind advice. We estimated that whether SP administration for 8 weeks enhances the muscle size and strength in 12-month-old male C57BL/6J mice, compared with 6-week-old male C57BL/6J mice. At that time, since appropriate concentration of SP administration was not decided yet, we provided to mice with 250 mg/kg/day of SP. According to our supplementary data, SP restores age-related declines in mass of hindlimb muscle. Also, age-related decreases in muscle strength were increased by SP. While body mass of OM+SP200 has no significant difference by SP treatment, it was slightly decreased compared OM.

Supplementary data 1. Effect of SP on muscle size and strength, and body weight in old mice. (A) Micro-CT image of hindlimb muscle. (B) Relative hindlimb muscle volumes were quantitated compared with those of 0 week. (C) Latency of forced swimming test was measured for 30 min. (D) Body weight of the mice after 8 weeks of experimental period. * p < 0.05, ** p < 0.01 compared to young mice group at 0 week; # p < 0.05, ## p < 0.01 compared to old mice group at 0 week.

Also, as your comment, skeletal muscles consist of various myosin heavy chain (MyHC). Therefore we investigated the enhancing effect of SP on muscle fiber area by using western blot and immunostaining. The antibody of MyHC used in this study was “Anti-heavy chain Myosin/MYH3 antibody (ab124205)” and it was purchased from Abcam. We revised manuscript as “In addition, H&E-stained images and MyHC immunostaining data of gastrocnemius muscle (Fig. 6B) show that SP treatment dose-dependently increased fiber cross-section area- which implies that SP encourages myogenic differentiation.” in result section.

Lastly, we explored the mitochondrial activity in gastrocnemius muscle of obese mice. Recent research reported that muscle mithochondial-related genes including NRF1, PGC1α and UCP3 regulates insulin resistance during obesity [5, 6]. Especially, PGC1α is well-known transcription factor correlated with sarcopenia and metabolic disease during aging [7, 8]. Therefore, our finding indicates that SP increased the expression level of these protein implies SP may increase the mitochondrial respiration in skeletal muscle in HFD-induced obese mice.

10-In figure 6, the authors claim that SP treatment increased fiber cross-section area.

If the authors have made analyses of the whole section area and quantified the number and surface area of the fibers it would be nice to show this.

 Answer) As your advice, we revised the Figure 6 with quantitative data of muscle fiver area (%). We also added the information in figure legend as “(B) Hematoxylin and eosin staining and immunofluorescence staining of MyHC in muscle from mice treated for 6 weeks. Its relative average area of 20-25 muscle fiber was quantified per same area using Image J software.”

11-Furthermore, the authors claim that SP treatment increased MyHC expression in HF diet. This statement is nonsense. The muscle remains muscle and it is mostly composed of MyHC. Their antibody is probably directed against a type of myosin and their western-blots suggest rather that SP induces a change in the contractile type.

Which isoform of MyHC recognizes your antibody?

 Answer) As we answered n Q6, skeletal muscles consist of various myosin heavy chain (MyHC). Therefore we investigated the enhancing effect of SP on muscle fiber area by using western blot and immunostaining. The antibody of MyHC used in this study was “Anti-heavy chain Myosin/MYH3 antibody (ab124205)” and it was purchased from Abcam. We revised as “In addition, H&E-stained images and MyHC immunostaining data of gastrocnemius muscle (Fig. 6B) show that SP treatment dose-dependently increased fiber cross-section area- which implies that SP encourages myogenic differentiation.” in result section. We appreciate that your suggestion

12-It is essential that you indicate the exact references of the antibodies used. For example, some phosphorylations activate AKT while others inhibit it. The interpretation is not the same...

 Answer) As reviewer’s comment, we added phosphorylation site of each factor in “2.1 Materials” on page 3 line 100-102, as “Anti-C/EBPα, PPARγ, myogenin, AKT, p-AKT (Ser 473), AMPK, and p-AMPK (Thr 172) antibodies were obtained from Cell Signaling Technology (Danvers, MS, USA). Anti-DGAT1, GAPDH, Fbx32, IRS, and p-IRS (Thr 632) antibodies were purchased from Santa Cruz Biotechnology, CA, USA.”

13-The authors claim that SP may prevent sarcopenia by enhancing muscle differentiation and inhibiting the muscle atrophy induced by HF diet.

Influence of SP on muscle Weight? Have you studied autophagy or synthesis flux? Have you studied the influence of SP on skeletal muscle regeneration?

 Answer) As we answered in Q9, we observed that SP prevents age-related sarcopenia by enhancing muscle enduarance. Although we could not measure the weight of each muscle of HFD-induced obese mice, our data indicates that SP may prevent sarcopenia by examing the expression levels of E3-ubiquitinase factors (MuRF1 and Fbx32), mitochondrial biogenesis related factors (UCP3, NRF1 and PGC1α), and myoblast differentiation factors (MyHC and Myogenin) in skeletal muscle of HFD fed mice. As your kind suggestion, even though we have to study autophagy or synthesis flux of muscle in further, please understand we was first time to study on muscle dystrophy. Reviewer’s kind advices will be helped us in setting our further research.

14-The authors claim that SP stimulates myoblast differentiation.

Did you measure the fusion index to state this?

Answer) As reviewer’s advice, we revised Figure 7 with fusion index (%). To estimate the percentage fusion, we calculated at the end of differentiation by dividing the number of nuclei within multinucleated myofibers by the total number of nuclei.

References

[1] Lee K, Jin H, Chei S, Lee J-Y, Oh H-J, Lee B-Y. Dietary Silk Peptide Prevents High-Fat Diet-Induced Obesity and Promotes Adipose Browning by Activating AMP-Activated Protein Kinase in Mice. Nutrients. 2020;12:201.

[2] Sunmin Park TZ, Jing Yi Qiu, Xuangao Wu, Boo Yong Lee   Silk amino acid consumption improves anti-diabetic symptoms by potentiating insulin secretion and preventing gut microbiome dysbiosis in non-obese type 2 diabetic animals. Nutrients stage of publication (in press).

[3] Lee S, Kim C, Kwon D, Kim MB, Hwang JK. Standardized Kaempferia parviflora Wall. ex Baker (Zingiberaceae) Extract Inhibits Fat Accumulation and Muscle Atrophy in ob/ob Mice. Evidence-based complementary and alternative medicine : eCAM. 2018;2018:8161042.

[4] Kim M-B, Kim T, Kim C, Hwang J-K. Standardized Kaempferia parviflora extract enhances exercise performance through activation of mitochondrial biogenesis. Journal of medicinal food. 2018;21:30-8.

[5] Devarshi P, McNabney S, Henagan T. Skeletal muscle nucleo-mitochondrial crosstalk in obesity and type 2 diabetes. International journal of molecular sciences. 2017;18:831.

[6] Kang C, Ji LL. Role of PGC-1α signaling in skeletal muscle health and disease. Annals of the New York Academy of Sciences. 2012;1271:110.

[7] Wenz T, Rossi SG, Rotundo RL, Spiegelman BM, Moraes CT. Increased muscle PGC-1α expression protects from sarcopenia and metabolic disease during aging. Proceedings of the National Academy of Sciences. 2009;106:20405-10.

[8] Wenz T. Mitochondria and PGC-1α in aging and age-associated diseases. Journal of aging research. 2011;2011.

Round 2

Reviewer 2 Report

In this revised version the authors have answered a number of questions, but I still have many comments.

Major criticisms:

1- Authors should indicate the exact antibody references.

2-The effect of SP on the body temperature of mice probably explains in part why these animals are resistant to obesity.

However, this raises many questions about SP administration. Is this hyperthermia related to infection? Does it induce inflammatory disease? Is it rather a decoupling effect of SP itself?

In any case, if SP treatment induces chronic hyperthermia there is a significant risk of neurological disorders, tachycardia and muscle problems.

It is imperative that this be discussed.

3- Similarly, the conclusion that "Thus, SP might prove to be a beneficial dietary supplement for patients with obesity and hyperglycemia in association with sarcopenia. " seems to me to be very very optimistic in view of hyperthermia induced by SP and the potential risks this could pose to patients.

This must be taken into account in your conclusions.

4-Using an antibody raised agisnt myosin MYH3 in adult mice makes no sense. It is an embryonic myosin that is only minimally expressed... In adult mice you have myosins type I, IIa, IIx and IIb.

Either you study the expression of these myosins to know if SP induces a change in the typology of muscle fibers or you remove the data concerning myosin MYH3.

5- To determine the cross-section area of the muscle fibers you measured 20 to 25 fibers using MYH3 immunofluorescence staining.

Again it doesn't make sense. To study CSAs in mice you must use an antibody raised against muscle membrane proteins such as dystrophin or laminin, and not an embryonic myosin. In addition, the gastrocnemius contains at least 10,000 fibers and is not a homogeneous muscle. Under these conditions, measuring 25 fibers out of 10 000 means nothing statistically. Either the authors analyze the CSAs with adequate staining and taking into account the whole slice or they remove this part.

6-The authors should include the additional figure 1 directly in the manuscript.

7-For the C2C12 study, MYH3 staining can be used because this myosin is expressed in the early stages of myoblast differentiation. The fusion index should be expressed as a percentage of differentiation and not as a percentage of the control.

7-The sentence in the discussion (line 556-557) “These results suggest that SP increases myoblast differentiation, which should improve muscle endurance capacity.” doesn't make sense. If SP stimulates myoblasts differentiation it rather suggests that SP could have a role on muscle regeneration but not on endurance capacities. Endurance capacity depends on many parameters such as the typology of muscle fibers and the activity of the mitochondrial respiratory chain. Same comment for lines 567-568.  

These sentences should be modified like the sentences relating to the expression of MYH3 in adult muscle.

8- I don't think there is a relationship between the expression of atrogenic factors and myoblast differentiation (line 564-567).

Degradation and protein synthesis in muscle are events subsequent to differentiation.

Minor criticisms:

1-line 138: it’s not oral glucose tolerance testing…

2- Supplementary Figure S1. Effect of SP on muscle size and strength, and body weight in 12-month-old mice.

It's important to specify the age. 12 months is old but the life expectancy of a C57Bl6 mouse is 24 months. So it's not very old either.

Author Response

Dear reviewer,

We appreciate to consider this paper for publication in Cells. We are very pleasure to have been given the opportunity to revise our manuscript, “Effect of Dietary Silk Peptide on Obesity, Hyperglycemia, and Skeletal Muscle Regeneration in High-fat Diet-Fed Mice”. We have addressed the reviewer’s comments point-by-point and made the necessary changes to the manuscript.

In second major revision, we made the following modifications based on the reviewer's recommendations: Explanation of the body temperature data was added with Table 2. Furthermore, we removed MYH3 data and added western data of MyoD expression in Figure 6 (in vivo data), according to reviewer’s kind comments. Regarding on it, we revised the discussion section implying that SP has a potential to treatment for obesity-related hyperglycemia and sarcopenia. Aslo, supplementary data was moved as Figure7 in main manuscript, and changed the term of old mice (OM) as adult mice (AD) in Figure 7. All revised descriptions were highlighted in yellow.

We hope that the manuscript is acceptable for publication in Cells, and declare that authors of this work have no conflict of interests. Thank you for your hard work.

Sincerely yours,

Boo-Yong Lee, Ph.D.

Second major revisions of Cells-702946 (Please see the attachment)

Reviewer2

1- Authors should indicate the exact antibody references.

 Answer) According to reviewer’s comment, we added antibody reference in 2.1. Materials on page 2-3; “Anti-Carnitine palmitoyltransferase 1 (CPT1, ab128568), GLUT4 (ab35826), MuRF1 (ab172479), MYH3 (ab124205), NRF1 (ab175932), PPARα (ab24509), PRDM16 (ab202344), Fbx32 (ab168372), and UCP3 (ab10985), were purchased from Abcam. Anti-PGC1α (sc13067), AKT (cs9272), p-AKT (Ser 473, cs9271), AMPK (cs2532), and p-AMPK (Thr 172, cs2535) antibodies were obtained from Cell Signaling Technology (Danvers, MS, USA). Anti-C/EBPα (sc61), PPARγ (sc7273), myogenin (sc12732), DGAT1 (sc32861), MyoD (sc760), GAPDH (sc365062), IRS (sc559), and p-IRS (Tyr 632, sc17196) antibodies were purchased from Santa Cruz Biotechnology (Santa cruz, CA, USA). “

2-The effect of SP on the body temperature of mice probably explains in part why these animals are resistant to obesity.

However, this raises many questions about SP administration. Is this hyperthermia related to infection? Does it induce inflammatory disease? Is it rather a decoupling effect of SP itself?

In any case, if SP treatment induces chronic hyperthermia there is a significant risk of neurological disorders, tachycardia and muscle problems.

It is imperative that this be discussed.

 Answer) We agree with you that acute infection or systemic inflammation is usually accompanied by changes in body temperature. Hyperthermia contributes to significant cellular damage and death in a manner that is dependent on both the amount and duration of temperature elevation [1]. Therefore, we investigated that effect of SP on serum creatinine, AST, and ALT of HFD-fed mice, by using commercial enzyme-linked immunosorbent assay (ELISA)/calorimetric assay kits. We used commercial alanineaminotransferase (ALT), aspartate aminotransferase (AST), and creatinine kits, and each absorbance was measured at appropriate wavelengths using a plate reader (BioTek Instruments Inc. Winooski, VT, USA). Creatinine is a marker of renal injury, and because muscle is the primary organ that synthesizes creatinine, serum creatinine levels are associated with decreased muscle mass [2]. Also, inflammation is considered to play a major role in the pathophysiology of hepatic damage, and can occur hyperthermia [3]. It is reported that abnormally elevating enzyme blood level of AST and ALT indicates hepatic cellular injury [4]. However, our findings indicated that SP has no significant differences in serum creatinine, AST and ALT concentration. Therefore, SP treatment does not induce acute kidney or liver injury. We added these contents as Table 2 in manuscript, as reviewer’s comment.

Moreover, we suggest that increasing of rectal temperature is due to uncoupling protein (UCP1) function in this study. UCP1 is transporter, present in the mitochondrial inner membrane, that mediate a regulated discharge of the proton gradient that is generated by the respiratory chain, so therefore heat generated [5]. Especially, UCP1 is predominantly expressed in BAT or Beige cells and plays a thermogenic role through the catalysis of proton-leak. Our previous study reported that SP promotes WAT-to-BAT differentiation in subcutaneous and visceral WAT by increasing the expression level of UCP1 [6].

Table 2. Effect of SP treatment on blood parameter in HFD-fed mice.

Group

Blood parameter (mg/dL)

CD

HFD

HFD+SP50 *

HFD+SP200 *

Creatinine

0.22 ± 0.04 a

0.25 ± 0.04 a

0.22 ± 0.03 a

0.23 ± 0.02 a

Aspartate aminotransferase (AST)

36.67 ± 7.37 a

43.33 ± 5.51 a

35.53 ± 9.07 a

35.33 ± 3.51 a

Alanineaminotransferase (ALT)

70.33 ± 9.50 a

76.00 ± 5.20 a

64.67 ± 12.42 a

62.67 ± 8.33 a

* (mg/kg/day), Data are expressed as mean ± SD (n = 6). Values with different letters are significantly different, p < 0.05 (a > b).

3- Similarly, the conclusion that "Thus, SP might prove to be a beneficial dietary supplement for patients with obesity and hyperglycemia in association with sarcopenia. " seems to me to be very very optimistic in view of hyperthermia induced by SP and the potential risks this could pose to patients.

This must be taken into account in your conclusions.

Answer) According to reviewer’s advice, we added the explanation in discussion section as we answered in Q2.; “To assess whether SP affect body temperature related to infection in mice, serum creatinine levels, AST and ALT were measured. Creatinine is a biomarker used to determine renal function, which is primary produced in muscle. Also liver function parameters including AST and ALT are representative factors of lipid damage, thus abnormally elevating enzyme blood levels indicates hepatic cellular injury. However, there was no significant difference in these serum concentrations of creatinine, ASL, and ALT among groups. Consequently, our data indicated that the effect of SP on kidney or liver injury may preclude.” In addition, we re-wrote the last sentence in conclusion as “Thus, SP might prove to be a beneficial dietary supplement for prevention of obesity and hyperglycemia in association with sarcopenia.”

4-Using an antibody raised agisnt myosin MYH3 in adult mice makes no sense. It is an embryonic myosin that is only minimally expressed... In adult mice you have myosins type I, IIa, IIx and IIb.

Either you study the expression of these myosins to know if SP induces a change in the typology of muscle fibers or you remove the data concerning myosin MYH3.

Answer) In this study, when mice were 11-week-old (after 6 weeks experimental period) we analyzed the expression of MYH3 using Myosin/MYH3 (ab124205) antibody. However, this protein is known to abundantly present in fetal skeletal muscle and not present or barely detectable in heart and adult skeletal muscle, as reviewer’s advice. Therefore, we decided to remove all of MYH3 data in Figure 6. Instead, we have analyzed the expression level of MyoD (sc-760) in muscle tissue. Regarding on it, we re-described in result 3.6; “Indeed, 200 mg/kg/day SP administration significantly increased myogenin to levels similar to those in CD-fed mice, and the MyoD expression in mice treated with 200 mg/kg/day SP was higher than that of CD-fed mice.”. We also revised discussion section; “The present study has shown that SP increases the expression of genes involved in skeletal muscle differentiation and reduces the expression of those involved in muscle degradation in mice. Specifically, HFD-induced obesity was associated with inhibition of the MyoD/Myogenin pathway in mice skeletal muscle. However, administration of SP increased the expression of these proteins in skeletal muscle. Recent studies reported that myogenesis is regulated by myogenic regulatory factors such as MyoD, Myogenin and MYHs [7]. MyoD is dispensable for skeletal muscle development and regulates the skeletal myogenic developmental program, which induces myogenic marker such as MYHs and Myogenin [8]. Also, Myogenin is required for myoblast fusion into myotubes. Our results suggest that SP treatment increases myoblast differentiation by myogenic factors such as MyoD, Myogenin and MYH3 in C2C12, so therefore SP may improve muscle regeneration.” We appreciated for your advice.

5- To determine the cross-section area of the muscle fibers you measured 20 to 25 fibers using MYH3 immunofluorescence staining.

Again it doesn't make sense. To study CSAs in mice you must use an antibody raised against muscle membrane proteins such as dystrophin or laminin, and not an embryonic myosin. In addition, the gastrocnemius contains at least 10,000 fibers and is not a homogeneous muscle. Under these conditions, measuring 25 fibers out of 10 000 means nothing statistically. Either the authors analyze the CSAs with adequate staining and taking into account the whole slice or they remove this part.

Answer) We removed immunostaining data of MYH3, and H&E data was presented in duplicate by using two different mice. Also, we calculated and revised the CSA data of muscle using H&E sections as Figure 6C. We wrote as “Quantification of the cross-sectional area (CSA) of muscle myofibers. The CSA of each muscle fiber in each field was measured using Image J program.” in Figure 6 legend.

Result 3.6. was revised as “In addition, H&E-stained images of muscle in hindlimb (Fig. 6B-C) is established that SP treatment dose-dependently increased the cross-sectional area (CSA), which implies that SP affects on size of muscle fibers.”

Discussion also revised as “Consistent with this, the present study has shown that the HFD-induced reduction in grip strength per unit body mass was ameliorated by SP. Indeed, HFD induced typical features of muscle wasting, such as lower fiber CSA value.”

6-The authors should include the additional figure 1 directly in the manuscript.

  Answer) According to reviewer’s advice, we moved the supplementary data 1 as Figure 7 in manuscript and revised the subtitle, like this; “3.6. SP increases the expression of genes determining skeletal muscle regeneration and reduces those involved in sarcopenia”. Furthermore, we re-named 12-month-old mice (OM) to adult mice (AM) as shown in below. Discussion was also revised; “Lastly, our last in vivo data showed that age-related loss of muscle mass and strength can be restore by impaired recovery of muscle mass following a treatment of SP in 12-month-old mice. Especially, SP increased hindlimb muscle so that the physical activity of the adult mice was increased. While body mass of AM+SP250 has no significant difference by SP treatment, it was slightly decreased compared AM. Collectively, SP treatment inhibited age-related reduction in muscle mass in adult mice, thereby muscle strength and physical function were recovered. Thus, SP supplement is considered to enhance physical strength as well as improve abnormal status among elderly. Therefore, SP may be useful as a food supplement to limit the development of sarcopenia alongside obesity or aged induced obesity.”

7-For the C2C12 study, MYH3 staining can be used because this myosin is expressed in the early stages of myoblast differentiation. The fusion index should be expressed as a percentage of differentiation and not as a percentage of the control.

Answer) We represented the fusion index of C2C12 as a percentage of differentiation in Fig. 8B. Moreover, we added the optical images of C2C12 cells in the presence or absence of SP in Fig. 8B. These images of C2C12 morphological change by SP treatment indicates that SP promoted the myoblast fusion during differentiation.

8-The sentence in the discussion (line 556-557) “These results suggest that SP increases myoblast differentiation, which should improve muscle endurance capacity.” doesn't make sense. If SP stimulates myoblasts differentiation it rather suggests that SP could have a role on muscle regeneration but not on endurance capacities. Endurance capacity depends on many parameters such as the typology of muscle fibers and the activity of the mitochondrial respiratory chain. Same comment for lines 567-568.  

These sentences should be modified like the sentences relating to the expression of MYH3 in adult muscle.

Answer) According to reviewer’s comments, we changed line 556-557 to like this: “Our results suggest that SP treatment increases myoblast differentiation by myogenic factors such as MyoD, Myogenin and MYH3 in C2C12, so therefore SP may improve muscle development”. In addition, lines 567-568 were also revised as “We found that the expression of Fbx32 and MuRF1 was lower in the muscle of SP-treated groups than in the HFD-fed group. Thus, our results indicate that SP may have the potential to prevent or reverse muscle atrophy by modulating the expression of these ubiquitine proteases.”

To sum up, our data showed that SP treatment promote muscle regeneration by enhancing the expressions of myogenic factors (MyoD, Myogenin, MYH) and mitochondrial biogenic factor (PGC1α, NRF1, UCP3), and reducing the expression of sarcopenic genes (MURF1, Fbx32) in HFD-fed mice. Putting together these results, we decided to change the title of this paper as “Effect of dietary silk peptide on obesity, hyperglycemia, and skeletal muscle regeneration in high-fat diet-fed mice”.

9- I don't think there is a relationship between the expression of atrogenic factors and myoblast differentiation (line 564-567).

Degradation and protein synthesis in muscle are events subsequent to differentiation.

Answer) We agree that degradation and protein synthesis in muscle are events subsequent to differentiation. Thus, for better logical deployment of our manuscript, we deleted the Western blot data of MuRF1 and Fbx32 of C2C12, as shown in Q7 answer. Also, we revised discussion; “To overcome sarcopenia, it is important to not only stimulate myogenesis, but also to inhibit muscle wasting. Despite the growning research on the treatment of sarcopenia, the molecular mechanisms that control obesity-related change in muscle mass are not fully understood. The E3-ligase F-box protein system is one of the major pathways that modulate muscle degradation, and it plays a vital role in regulating muscle mass. Specifically, activation of the ubiquitin protease pathway in muscle atrophy is associated with increases in expression of two muscle-specific proteins, Fbx32 and MuRF1 [9]. Furthermore, HFD consumption causes skeletal muscle to overexpress these proteins. In present study, SP treatment reduced the expression level of those involved in muscle degradation in HFD-fed obese mice. We found that the expression of Fbx32 and MuRF1 was lower in the muscle of SP-treated groups than in the HFD-fed group. Thus, our results indicate that SP may has the potential to prevent or reverse muscle atrophy by modulating the expression of these ubiquitine proteases.”

Minor criticisms:

1-line 138: it’s not oral glucose tolerance testing…

  Answer) Thank you for your advice, we revised 2.5 as “Measurement of rectal temperature”

2- Supplementary Figure S1. Effect of SP on muscle size and strength, and body weight in 12-month-old mice.

It's important to specify the age. 12 months is old but the life expectancy of a C57Bl6 mouse is 24 months. So it's not very old either.

 Answer) As your advice, we revised (old mice) OM to (Adult mice) AM. In recent studies regarding on sarcopenia have been explored in vivo experiment by using 18-24-month-old mice as old mice group [10,11]. Although we could not analyze the muscle metabolism using old mice due to financial problem, we used the adult mice as a middle aged group and administrated of SP for 8 weeks.

<References>

Halpin, L.E.; Gunning III, W.T.; Yamamoto, B.K. Methamphetamine causes acute hyperthermia‐dependent liver damage. Pharmacology research & perspectives 2013, 1. Kaysen, G.A. Biochemistry and biomarkers of inflamed patients: why look, what to assess. Clinical Journal of the American Society of Nephrology 2009, 4, S56-S63. Akcay, A.; Nguyen, Q.; Edelstein, C.L. Mediators of inflammation in acute kidney injury. Mediators of inflammation 2009, 2009. Kim, S.E. Optimal Evaluation of the Results of Liver Function Tests. Korean J Med 2019, 94, 89. Ledesma, A.; de Lacoba, M.G.; Rial, E. The mitochondrial uncoupling proteins. Genome biology 2002, 3, reviews3015. 3011. Lee, K.; Jin, H.; Chei, S.; Lee, J.-Y.; Oh, H.-J.; Lee, B.-Y. Dietary Silk Peptide Prevents High-Fat Diet-Induced Obesity and Promotes Adipose Browning by Activating AMP-Activated Protein Kinase in Mice. Nutrients 2020, 12, 201, doi:10.3390/nu12010201. Charge, S.B.; Rudnicki, M.A. Cellular and molecular regulation of muscle regeneration. Physiological reviews 2004, 84, 209-238. Megeney, L.A.; Kablar, B.; Garrett, K.; Anderson, J.E.; Rudnicki, M.A. MyoD is required for myogenic stem cell function in adult skeletal muscle. Genes & development 1996, 10, 1173-1183. Abrigo, J.; Rivera, J.C.; Aravena, J.; Cabrera, D.; Simon, F.; Ezquer, F.; Ezquer, M.; Cabello-Verrugio, C. High fat diet-induced skeletal muscle wasting is decreased by mesenchymal stem cells administration: implications on oxidative stress, ubiquitin proteasome pathway activation, and myonuclear apoptosis. Oxidative medicine and cellular longevity 2016, 2016. Uchitomi, R.; Hatazawa, Y.; Senoo, N.; Yoshioka, K.; Fujita, M.; Shimizu, T.; Miura, S.; Ono, Y.; Kamei, Y. Metabolomic analysis of skeletal muscle in aged mice. Scientific reports 2019, 9, 1-11. Altun, M.; Besche, H.C.; Overkleeft, H.S.; Piccirillo, R.; Edelmann, M.J.; Kessler, B.M.; Goldberg, A.L.; Ulfhake, B. Muscle wasting in aged, sarcopenic rats is associated with enhanced activity of the ubiquitin proteasome pathway. Journal of Biological Chemistry 2010, 285, 39597-39608.
